# Mutational landscape of triple-negative breast cancer in African American women

Song Yao [1,13] ✉, Lei Wei [2,13], Qiang Hu [2,13], Song Liu[2,13], Zarko Manojlovic [3], Peter N. Fiorica [1], Mark Long [2], Gary R. Zirpoli [4], Qiuyin Cai [5], Jirong Long [5], Jie Ping [5], Mollie E. Barnard [4], Yuxin Jin [6], Mitsuko Murakami[2], Jianmin Wang [2], Qianqian Zhu [2], Warren Davis[1], Jianhong Chen[1], Rochelle P. Ondracek[1], Thaer Khoury[7], Shipra Gandhi[8], Kazuaki Takabe[9], Naomi Ko [10], Maureen Sanderson[11], Chi-Chen Hong[1], Elisa V. Bandera[12], David W. Craig[6], Christine B. Ambrosone [1], Julie R. Palmer[4], Wei Zheng[5] & John D. Carpten[6]

African American (AA) women have the highest incidence of triple-negative breast cancer (TNBC) among all ancestral groups, but are underrepresented in cancer genomic studies. In 462 AA women with TNBC, we characterized the tumor mutational landscape by whole-exome sequencing and RNA sequencing. We unveiled a high-resolution mutational portrait of TNBC in AA women reminiscent of that in Chinese and non-Hispanic white women, with no evidence of associations of mutational features with African ancestry. We also made some distinctive discoveries, including an almost complete dominance of *TP53* mutations, low frequency of *PIK3CA* mutations and mutational signature-based subtypes with etiologic and prognostic significance. These findings do not support major ancestral differences in TNBC biology at the level of somatic mutations. Our study contributes considerably to diversifying the knowledge base of breast cancer genomics and provides insights into the disease etiology, disparities and therapeutic vulnerability of TNBC in AA women.

Triple-negative breast cancer (TNBC) remains an enigma in oncology and epidemiology. This subtype is defined by the lack of expression of estrogen receptor (ER), progesterone receptor (PR) and ErbB2 on immunostaining and represents 10–15% of all breast cancer diagnoses[1]. Although new treatments such as PARP inhibitors and immune checkpoint inhibitors improve outcomes for some patients[2], TNBC is still the most deadly subtype of breast cancer. A prominent epidemiological feature of TNBC in the United States is the disproportionately high incidence in African American (AA) women, estimated at 25.2 per 100,000 in comparison to 12.9 per 100,000 in non-Hispanic white (NHW) women[3].

One important question regarding the distinctive TNBC epidemiology in AA women is whether it is a result of differences in cancer biology or socio-environmental exposures[1]. In recognition of TNBC as a group of heterogeneous diseases, multi-omic breast tumor characterization is warranted. However, despite a high disease burden, AA women have long been underrepresented in cancer genomic studies[4]. Of the few breast tumor sequencing studies in AA women[5–8], none focused on

[1]Department of Cancer Prevention and Control, Roswell Park Comprehensive Cancer Center, Buffalo, NY, USA. [2]Department of Biostatistics and Bioinformatics, Roswell Park Comprehensive Cancer Center, Buffalo, NY, USA. [3]Department of Translational Genomics, Keck School of Medicine of USC, Los Angeles, CA, USA. [4]Slone Epidemiology Center, Boston University, Boston, MA, USA. [5]Division of Epidemiology, Department of Medicine, Vanderbilt Epidemiology Center, Vanderbilt-Ingram Cancer Center, Vanderbilt University Medical Center, Nashville, TN, USA. [6]Comprehensive Cancer Center, City of Hope, Duarte, CA, USA. [7]Department of Pathology, Roswell Park Comprehensive Cancer Center, Buffalo, NY, USA. [8]Department of Medicine, Roswell Park Comprehensive Cancer Center, Buffalo, NY, USA. [9]Department of Surgical Oncology, Roswell Park Comprehensive Cancer Center, Buffalo, NY, USA. [10]Department of Medical Oncology, Boston Medical Center, Boston, MA, USA. [11]Department of Family and Community Medicine, Meharry Medical College, Nashville, TN, USA. [12]Section of Cancer Epidemiology and Health Outcomes, Rutgers Cancer Institute of New Jersey, New Brunswick, NJ, USA. [13]These authors contributed equally: Song Yao, Lei Wei, Qiang Hu, Song Liu. ✉e-mail: song.yao@roswellpark.org

TNBC, except for one with 51 patients[9], in contrast to much larger such studies in Chinese and NHW women[10–12].

Herein, we assembled matched tumor and normal samples from self-identified AA women with TNBC in five population-based studies and carried out whole-exome sequencing (WES) and RNA sequencing analysis, with goals to chart the mutational landscape of TNBC in AA women and to investigate the implications for cancer etiology and therapy.

## Results

### Patient population

Paired tumor and normal samples from 513 self-identified AA women with TNBC were interrogated by WES. After data processing and quality control steps, 462 (90%) cases were included in the final analysis. Patient descriptive characteristics are shown in Supplementary Table 1. The average (±s.d.) age at diagnosis was 53 (±11) years, with 38% before the age of 50 years.

### Mutational landscape of TNBC in AA women

From the 462 tumors, we identified 39,103 mutations in the coding regions, including 36,059 (92%) single-nucleotide variants (SNVs) and 2,690 (7%) insertion/deletions (indels; Supplementary Fig. 1). The median mutation burden was 1.29 (range = 0.07–22.2) SNVs per Mb, with five tumors (1%) considered hypermutated (>10 SNVs per Mb)[13] (Supplementary Fig. 2), three of which carrying a mutation in mismatch repair genes (MLH1, MSH3 and LIG1). At the gene level, we identified nonsilent mutations in 11,273 genes (Supplementary Table 2), with a median of 47 (range = 1–664) mutated genes per tumor.

Figure 1a illustrates the compendium of somatic mutations in TNBC from AA women. The mutational landscape is predominated by alterations in TP53, with a total of 463 mutations found in 437 (95%) tumors, including 18 with two or more mutations. A majority (59%) of the mutations were recurrent and all but six were nonsilent. We classified 294 (66%) of the TP53 coding mutations as loss of function, 113 (26%) as gain of function, 2 (0.5%) as benign, and 33 (7%) as function unknown (most in-frame indels). Most tumors (n = 431 or 93%) had at least one nonsilent mutation. One tumor harbored E224E, a known cancer-driving synonymous mutation[14]. In addition, five tumors harbored intronic mutations only. Using transcriptomic data available from four of these tumors, we found evidence of aberrant RNA splicing in three (Extended Data Fig. 1).

### Confirmation of TP53 mutations

In validation analysis of TP53 mutations using transcriptomic data available from 260 patients, 215 (83%) of the 259 mutations identified by WES were detected at the RNA level (Extended Data Fig. 2). Given the uneven coverage of RNA sequencing, we restricted the analysis to tumors with ≥10× and were able to confirm 183 of 187 (98%) mutations in these tumors. The concordance reached 100% (109/109 mutations) in tumors with ≥30× coverage.

Furthermore, we resequenced the TP53 region in 338 tumors with DNA available (317 with and 21 without mutations) using targeted amplicon sequencing (TAS). These 317 tumors harbored 326 mutations identified by WES. We confirmed 324 (99%) mutations, with only two going undetected by TAS (Extended Data Fig. 2). Of the 21 tumors that had no TP53 mutations in WES data, TAS analysis identified two new mutations at low variant allele frequency (<5%).

### Known cancer genes and significantly mutated genes

Of the 11,273 genes harboring nonsilent mutations, 218 had a frequency ≥2% (≥10 tumors; Supplementary Table 2). Aside from TP53, all other genes were mutated at a much lower frequency (Fig. 1a). These included 16 known TNBC genes, namely, NOTCH1 (7%), RB1 (7%), KMT2D (6%), PIK3CA (5%), PTEN (5%), KMT2C (5%), BRCA1 (5%), NF1 (5%), SPEN (4%), FAT3 (4%), CREBBP (4%), PIK3R1 (3%), NOTCH2 (3%), BRCA2 (2%), ERBB2 (2%) and KDM6A (2%), and 7 known breast cancer genes, albeit not

specific to TNBC, ARID1A (4%), CIC (4%), GNAS (4%), AXIN1 (4%), RYR2 (3%), USH2A (3%) and GATA3 (2%). Additionally, 21 driver genes identified in previous pan-cancer analysis[15,16] mutated in ≥2% in our cohort. These included several well-known cancer genes or their family members with previously less recognized role in breast cancer—FGFR3, FGFR4, NOTCH3, KMT2B, EP300, FLT4 and FAT1.

Significantly mutated genes (SMG) analysis identified 13 genes with q ≤ 0.20 by two or more programs used (MutSigCV[17], MutSig2CV[18] and MuSiC[19]), which were all known cancer genes in TNBC (Supplementary Tables 3–6).

In comparisons across TNBC transcriptional subtypes[20,21], the luminal androgen receptor subtype had enrichment of somatic mutations in PTEN (P = 0.003) and PIK3R1 (P = 0.04) and slight depletion of TP53 (P = 0.009) mutation, consistent with previous studies in Chinese and NHW patients (Supplementary Fig. 3)[10,22,23].

### Pathogenic germline mutations

Using sequencing data from matched normal samples, we identified 124 germline mutations in nine known TNBC predisposition genes from 241 patients (Fig. 1a). Of these variants, 115 were found in gnomAD and other reference datasets[24–27], 22 being exclusive to populations of African ancestry (Supplementary Table 7). When minor allele frequency was compared with reference populations, 60 variants had a higher frequency in TNBC patients (P < 0.05; Extended Data Fig. 3). These results confirmed benignity for 28 of 30 variants classified as 'benign/likely benign' and pathogenicity for 23 of 25 variants classified as 'pathogenic/likely pathogenic' by ClinVar[28], while yielding new evidence of pathogenicity for 14 of 35 variants annotated as 'conflicting classification of pathogenicity' and 15 of 18 variants annotated as 'uncertain significance'. Moreover, we identified six variants, including two in PTEN and four in BRCA1, with no pathogenicity annotation in ClinVar[28], all of which had a higher frequency in TNBC cases than in the reference datasets, including R119C mutation in PTEN (P = 9 × 10⁻⁷). Lastly, we discovered nine new germline mutations not previously reported in any reference databases, including three in BRCA1, three in BRCA2, two in PALB2 and one in NF1. Two of the BRCA1 variants were deemed damaging in saturation genome editing[29].

### Copy number aberrations

Figure 1b illustrates the copy number abberation (CNA) landscape based on WES data captured with additional baits representing human array comparative genomic hybridization (aCGH) probes. We identified multiple substantial copy number gains or losses, including 18 regions containing known cancer genes (Supplementary Fig. 4). As expected, high-level MYC amplification was one of the most common copy number changes found in 36% of the tumors (Fig. 1c). Other known cancer genes residing within substantial aberrant regions defined by Genomic Identification of Significant Targets in Cancer (GISTIC2) included high-level amplifications of MCL1 (44%), AKT3 (21%), GATA3 (19%), E2F3 (18%), NFIB (13%), CCNE1 (12%), IRS2 (12%), PIK3CA (12%), MYB (8%), NOTCH2 (8%), EGFR (5%), FGFR2 (5%) and TERT (4%) and homozygous deletion of RB1 (8%), PTEN (6%), CDKN2A/CDKN2B (5%) and ESR1 (3%).

### Commonly altered signaling pathways

In gene set enrichment analysis (GSEA) based on 218 cancer driver genes harboring recurrent (≥1%) somatic mutations in the cohort, we identified four commonly altered signaling pathways (q < 0.05). The p53 signaling pathway was altered in most (95%), if not all, TNBC in our cohort (Fig. 2a). Mutations in other genes in the core p53 signaling pathway were much rarer (1% in ATM and 5% in CHEK2) and all co-occurred with TP53 mutations. When CNAs were also considered, the p53 signaling pathway was implicated in 97% (n = 450) of the tumors.

NOTCH1 was the second commonly mutated gene (7%) in our cohort. Nonsynonymous mutations were found at a lower frequency in three other NOTCH family genes, with an aggregated mutation

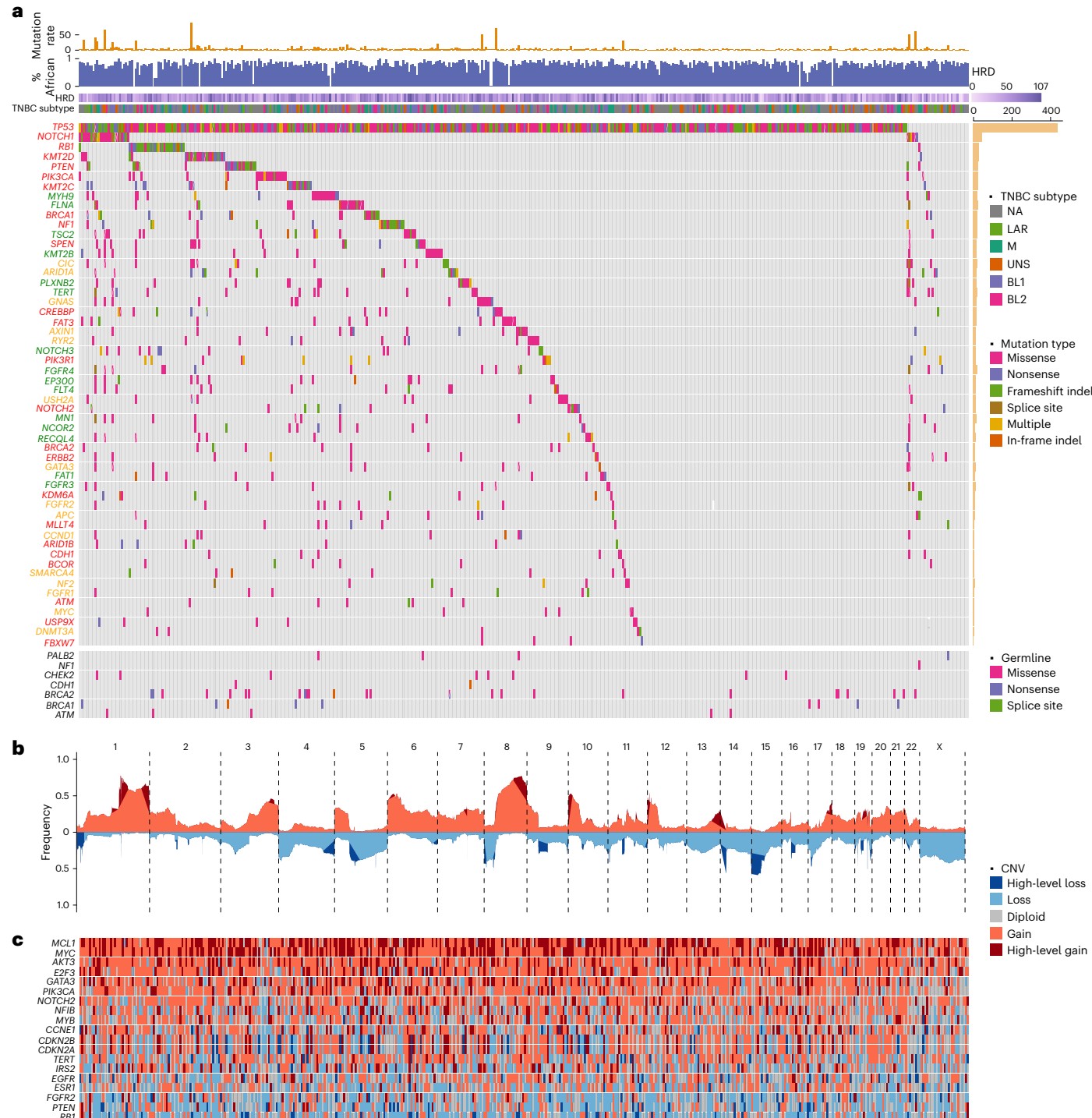

**Fig. 1 | Mutational landscape of TNBC from AA women. a**, CoMut plot of somatic and germline mutations in TNBC from AA women. Mutation rate is presented as the number of SNVs per Mb. The proportion of African ancestry was estimated based on germline variant data from matched normal DNA samples and presented as a numeric value between 0 and 1. TNBC subtype was classified based on tumor transcriptomic data available from 260 cases using the method discussed in ref. 20. HRD was estimated based on WES data using scarHRD R package[57]. Somatic mutations are sorted by mutation frequency and shown in the upper section of the CoMut plot. Germline variants are shown in the lower section of the plot. Gene symbols are labeled in colors to indicate known TNBC genes (red), breast cancer genes (yellow) and pan-cancer genes (green). NA, not available. **b**, Frequency of CNAs (*y* axis) across chromosomes (*x* axis), with red color for copy number gain, blue color for copy number loss and dark red and dark blue for regions tested substantially in GISTIC2 analysis. CNV, copy number variation. **c**, Heatmap of known cancer genes in substantial peaks identified by GISTIC2. BL1, basal-like 1; BL2, basal-like 2; LAR, luminal androgen receptor; M, mesenchymal; UNS, unassigned.

frequency of 14%, which was further increased to 22% when other genes in the core Notch signaling pathway were considered (Fig. 2b). Moreover, high-level amplifications were also identified in *NOTCH* family genes and two co-activators *EP300* and *CREBBP*.

*RB1* is the third most commonly mutated gene (7%) in our cohort, and homozygous deletion was observed in 8% of the tumors (Fig. 2c). In addition to germline mutations, *BRCA1* and *BRCA2* somatic mutations were observed in another 5% and 2% of tumors, respectively. Further,

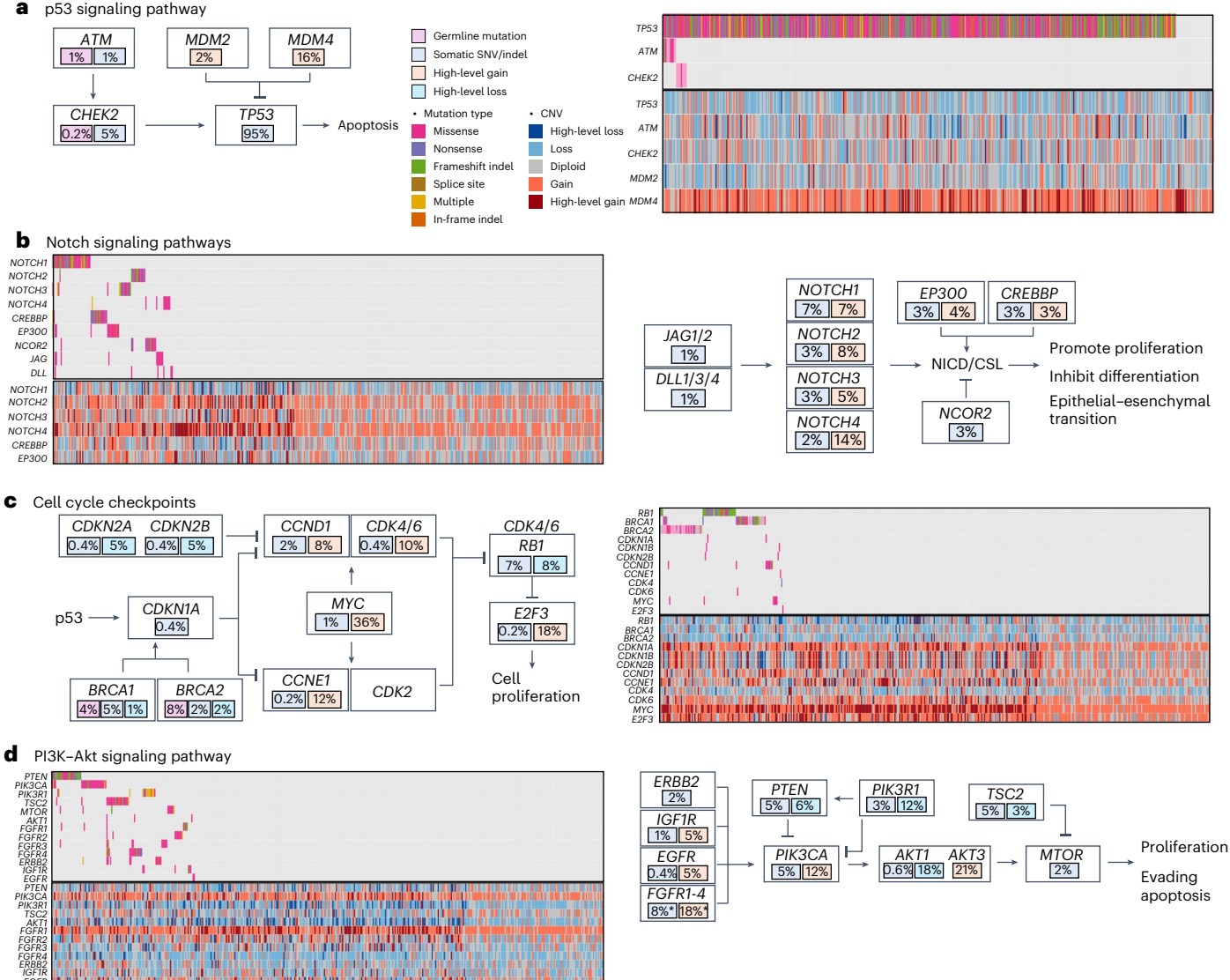

**Fig. 2 | Common signaling pathways altered in TNBC from AA women.**
Commonly altered pathways in TNBC from AA women based on GSEA results
using point mutations and indels. The numbers in the plots indicate the
percentages of tumors harboring the alterations that are color coded.
**a**, p53 signaling pathway. **b**, Notch signaling pathway. **c**, Cell cycle checkpoints.
**d**, PI3K–Akt signaling pathway.

5% of the tumors demonstrated homozygous deletion of *CDKN2A* and
*CDKN2B*. On the contrary, high-level amplification occurred at a high
frequency in several key genes driving cell cycle progression, including
*CCND1* (8%), *CCNE1* (12%), *CDK4*/*CDK6* (10%), *E2F3* (18%) and *MYC* (36%).

Several genes in the PI3K–Akt signaling pathway that encode
growth factor receptors demonstrated mutation and/or high-level
amplification, including the four *FGFR* family members, *ERBB2*, *IGFR1*
and *EGFR* (Fig. 2d). In addition, several core PI3K pathway members
were among the top mutated genes, including *PIK3CA*, *PTEN* and *TSC2*
each at 5%, plus *PIK3R1* at 3% and *MTOR* at 2%.

### Mutational signatures

Three de novo single base substitution (SBS) mutational signatures
were extracted from 457 tumors after excluding five hypermutated
samples, which were then decomposed to five of the COSMIC SBS96
signatures (Fig. 3a and Supplementary Fig. 5)[30]. These include two
clock-like signatures, SBS1 and SBS5, homologous recombination
deficiency (HRD)-related signature, SBS3, and two APOBEC-related
signatures, SBS2 and SBS13, all of which have previously been found

in TNBC in Chinese and NHW women[10,11,31,32]. The clock-like SBS1 and
SBS5 were moderately correlated with each other ($r = 0.45$, $P < 0.001$)
and together showed a moderate correlation with patient age ($r = 0.21$,
$P < 0.001$; Supplementary Fig. 6a).

While SBS1 and SBS5 were found in virtually all tumors, represent-
ing the dominant mutagenic processes in almost half of the tumors, the
HRD-related SBS3 dominated the other half (53%; Fig. 3a). As expected,
SBS3 was correlated with HRD score ($r = 0.62$, $P < 0.001$; Supplemen-
tary Fig. 6b), and was more active among patients carrying *BRCA1* and
*BRCA2* germline variants ($P = 0.005$; Supplementary Fig. 6c). Notably,
there was also a moderate negative correlation between SBS3 and SBS5
($r = -0.29$, $P < 0.001$). The two APOBEC-related signatures, SBS2 and
SBS13, were strongly correlated with each other ($r = 0.85$, $P < 0.001$)
and manifested in 35% TNBC tumors at lower activity relative to the
other three signatures.

For indels, three de novo signatures were extracted from 439
tumors. Decomposition analyses yielded six indel signatures in refer-
ence to COSMIC ID83 (ref. 30), including the following five known ones:
ID2 related to slippage during DNA replication, ID4 with no known

etiology, ID6 related to HRD, ID7 related to defective DNA mismatch damage repair and ID8 related to double strand break repair (Fig. 3a and Supplementary Fig. 5)[30]. ID6 and ID8 were the most active mutagenic processes found in 38% and 42%, respectively, of the tumors and both showed a moderate correlation with estimated HRD score ($r = 0.48$ and $r = 0.42$, respectively, $P < 0.001$; Supplementary Fig. 6d,e). The sixth indel signature, characterized by longer indels ≥5 bp was new and presented in one-third of the TNBC tumors, which displayed a weak negative correlation with HRD ($r = -0.21$, $P < 0.001$; Supplementary Fig. 6f).

When examined across TNBC transcriptional subtypes[20,21], the M subtype had relatively lower APOBEC-related SBS2 ($P = 0.005$) and SBS13 ($P = 0.09$) signatures, and the luminal androgen receptor subtype had lower HRD-related SBS3 ($P = 0.006$) but higher ID4 signature ($P = 0.006$; Supplementary Fig. 3).

## Genomic and immune differences by SBS subtype

We defined SBS signature-based TNBC subtypes by combining SBS1 + SBS5 (aging) and SBS3 (HRD). The differences between subtype 1 (low aging and high HRD) and subtype 3 (high aging and low HRD) were the most apparent, whereas subtype 2 (low aging and low HRD) and subtype 4 (high aging and high HRD) were somewhere in between. Tumor classified as subtype 1 had higher mutation rate, HRD score and pathological tumor infiltrating lymphocyte score, lower *BRCA1* expression and were less likely from older patients or those with higher body mass index (BMI; $P \le 0.05$; Fig. 3b). For every 10-year increment of age and every 5 kg m$^{-2}$ increment of BMI, the odds of having subtype 1 versus subtype 3 TNBC decreased by 68% ($P = 0.0001$) and 31% ($P = 0.008$), respectively.

Moreover, these two SBS subtypes differed in somatic mutations in several cancer driver genes, including higher mutation frequency of *ERBB2*, *GATA3* and *FGFR4*, and lower frequency of *DMD*, *INHBA*, *OGDHL*, *PLEKHG5* and *RYR2* in subtype 3 than in subtype 1 ($P < 0.05$; Fig. 3a). In addition, subtype 1 tumors were also more likely to have high-level amplification of *MYC*, *MCL1*, *AKT3*, *E2F3* and *GATA3* ($P < 0.05$). In analysis of gene expression-based immune signatures, subtype 1 manifested with stronger immune cytolytic activity signature[33] ($P = 0.002$) and two B cell signatures[34,35] ($P = 0.005$ and $P = 0.04$) than subtype 3 (Fig. 3c). Consistent with this, GSEA showed substantial enrichment of many immune response gene sets in subtype 1 relative to subtype 3 tumors (Supplementary Fig. 7).

## Mutational signatures and patient survival

As shown in Fig. 3d, higher SBS1 + SBS5 (aging) was associated with higher all-cause mortality (high versus low—hazard ratio (HR) = 1.97, 95% confidence interval (CI) = 1.24–3.13, $P = 0.004$), whereas higher SBS3 (HRD) was associated with lower mortality (HR = 0.55, 95% CI = 0.33–0.92, $P = 0.02$). The associations became only borderline substantial after adjusting for age, study and stage (Supplementary Table 8). No substantial association of patient survival was found with APOPEC signatures SBS2 or SBS13. In analyses of SBS signature-based TNBC subtype, patients with subtype 3 had the higher all-cause mortality, in comparison to those with SBS subtype 1 (HR = 2.63, 95% CI = 1.47–4.69, $P = 0.001$; Fig. 3d), which remained substantial after controlling for age

and cancer stage (HR = 1.96, 95% CI = 1.05–3.64, $P = 0.03$). Meta-analyses across the three studies show similar results (Extended Data Fig. 4). No substantial association was observed with subtype 2 or subtype 4.

## Comparisons of somatic mutations across ancestral groups

Figure 4 shows the three-way comparisons of mutation frequency of known breast cancer genes across AA patients from 'Breast Cancer in African Americans: Understanding Somatic Mutations and Etiology' (B-CAUSE) study ($n = 462$), Chinese patients from Fudan University Shanghai Cancer Center (FUSCC; $n = 279$)[10] and NHW patients ($n = 626$) pooled from The Cancer Genome Atlas (TCGA)[36], Sweden Cancerome Analysis Network—Breast (SCAN-B)[11] and Molecular Taxonomy of Breast Cancer International Consortium (METABRIC)[37], with the full results provided in Supplementary Table 9. The mutation frequencies were largely similar between Chinese and NHW patients, yet several genes had notably different mutation frequency from these in AA patients, including higher frequencies of *TP53* (95%, 78% and 75% in AA, Chinese and NHW patients, respectively; $P < 1 \times 10^{-9}$) and *NOTCH1* (7%, 2% and 4%, respectively; $P < 0.01$) mutations. On the contrary, a number of genes were found to mutate at a lower frequency in AA patients than in Chinese or NHW patients, most notably *PIK3CA* (5%, 19% and 15%, respectively; $P < 2 \times 10^{-6}$), *RYR2* (3%, 7% and 8%, respectively; $P < 0.03$) and *USH2A* (3%, 1% and 8%, respectively; $P < 9 \times 10^{-4}$), while the mutation frequency of *AKT1*, *ATR*, ATRX, *MAP3K1*, *PREX2* and *SETD2* were very low in AA patients (<1%) but relatively common (≥3%) in Chinese and/or NHW patients.

For *TP53* and *PIK3CA*, the two genes showing the largest mutation frequency discrepancy across the three patient populations, the gene mutation spectrums were, nevertheless, largely similar, with some minor yet notable differences. Most of the mutations in *TP53* were found in the DNA-binding domain (DBD), featured prominently with four hotspot mutations (R175, R213, R248 and R273), plus another hotspot mutation R342 in the tetramerization domain (Fig. 5a). Tumors from AA women had two other hotspot mutations, H179 and E286 in the DBD, which were absent in Chinese patients and at only low frequency in NHW patients. On the contrary, nonsense mutation R196* was rare in AA patients but more common in Chinese and NHW patients. The spectrum of *PIK3CA* mutations was dominated by one hyperactivating mutation, H1047R/L, in all three populations; however, the other three hyperactivating hotspot mutations, N345K, E542K and E545K, were found only in tumors from Chinese and NWH women but not from AA women (Fig. 5b).

## African ancestry and somatic mutational features

The median proportion of African ancestry was 0.82 (interquartile range = 0.74–0.89; Supplementary Fig. 8). There was no correlation of African ancestry with mutation rate, HRD score or any of the mutational signatures (Supplementary Fig. 9). There were also no differences in percent African ancestry by TNBC subtype or *TP53* hotspot mutations.

## Neoantigen analysis

Supplementary Fig. 10 shows the number of predicted neoantigens in each tumor (median = 2, range = 0–35) with the number of

---

**Fig. 3 | Mutational signatures in TNBC from AA women. a**, From top to bottom: the first row shows HRD estimated based on WES data using scarHRD R package[57]. The second row shows the proportion of African ancestry was estimated based on germline variant data from matched normal DNA samples and presented as a numeric value between 0 and 1. The third row shows TNBC subtype classified based on tumor transcriptomic data available from 260 cases using the method discussed in ref. 20. The fourth and fifth rows show SBS and ID (indel) mutational signatures, respectively. The sixth and seventh rows show genes with differences in frequency of somatic mutations and CNAs between SBS subtype 1 (low aging and high HRD) and subtype 3 (high aging and low HRD), respectively. **b**, Demographic and mutational features that show substantial differences between SBS subtype 1 ($n = 103$) and subtype 3 ($n = 131$) by the Wilcoxon test

and the *P* values were two-sided without adjustment for multiple comparisons. Subtypes 2 and 4 have characteristics that fall somewhere between subtypes 1 and 3 and are not shown. The bar in the middle of a box indicates the subgroup median, and the lower and upper edges indicate the first and third quartiles, respectively. The whiskers indicate the range in each subgroup. *P* values were derived from two-sided Wilcoxon test between Black and white patients. TILs, tumour-infiltrating lymphocytes. **c**, Tumor microenvironment immune signatures that show substantial differences between SBS subtype 1 and subtype 3 by the Wilcoxon test and the *P* values were two-sided without adjustment for multiple comparisons. **d**, Kaplan–Meier curves of all-cause mortality (death due to any cause) by SBS signatures, with *P* values derived from the log-rank test.

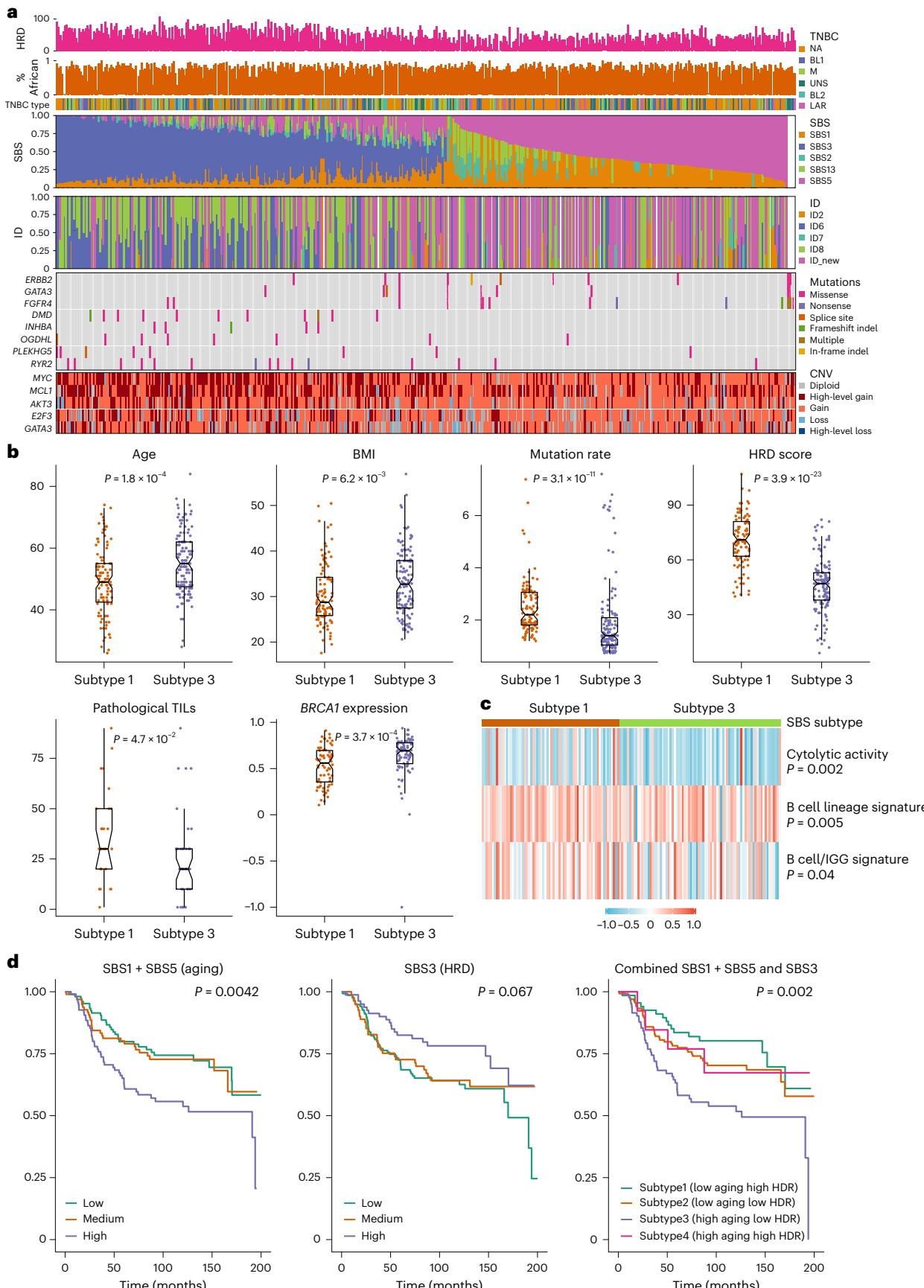

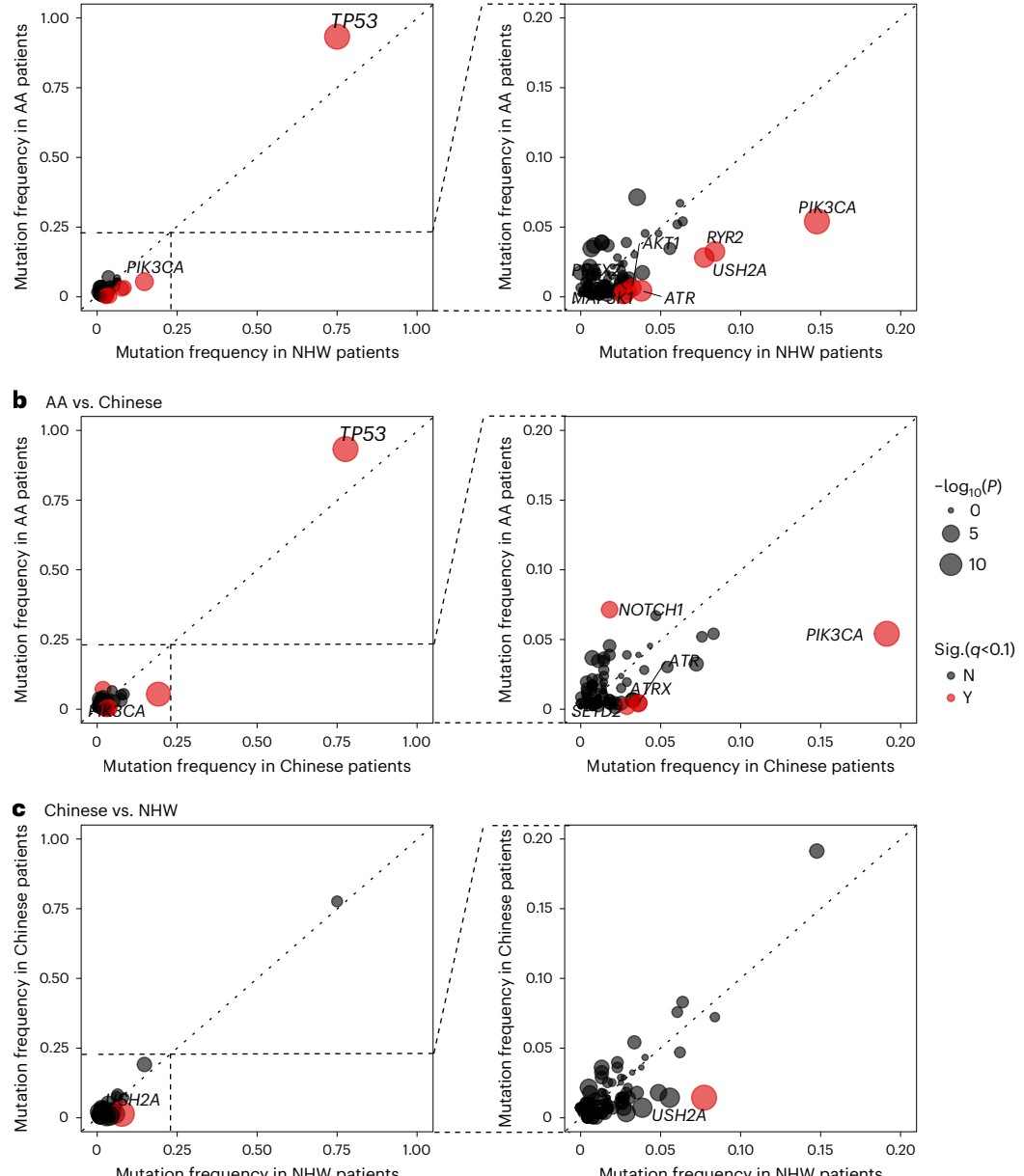

**Fig. 4 | Comparison of somatic mutations in TNBC between AA, Chinese and NHW women. a–c**, The frequency of nonsilent mutations in known breast cancer genes in TNBC from AA women (*y* axis) and NHW women (*x* axis) (**a**), AA women (*y* axis) and Chinese women (*x* axis) (**b**) and Chinese women (*y* axis) versus NHW women (*x* axis) (**c**). AA women were from B-CAUSE study; Chinese women were from FUSCC and NHW women were pooled from TCGA, SCAN-B and METABRIC. Each dot represents one gene with the dot size corresponding to the negated $\log_{10}$-transformed two-sided *P* value from comparison test. Genes that were substantial at *q* < 0.10 after FDR correction are shown in red.

nonsynonymous missense mutations, where moderate correlation was found between the two ($r$ = 0.58, $P$ < 0.001).

## RNA fusion events

We characterized fusion events in 260 TNBC patients with transcriptomic data. Using stringent filtering criteria, we identified 471 fusion mutations in 148 (56%) of the tumors, including seven recurrent fusions and 96 fusions involving a known cancer gene (Supplementary Table 10). The most common recurrent fusions were characterized by adjacent rearrangements involving *PTK2* or *ETV6*, the latter of which is a tumor suppressor that turns to an oncogene in its fusion forms[38]. We identified one tumor with *BCL2L14–ETV6* associated with mesenchymal TNBC[39] and another with *ETV6–NTRK3* that was a marker of secretory breast carcinoma, a rare basal-like breast cancer[40,41]. Six tumors

had fusions involving *PTK2* with multiple partners, and none retained the kinase domain (Supplementary Fig. 11). Moreover, three tumors contained *PARG–BMS1* fusion associated with metaplastic TNBC[42]. In addition, four tumors had fusion mutations involving *NOTCH2* or *NOTCH2NL*[43].

## Potential therapeutic targets in TNBC

Based on deleterious mutations in *BRCA1* and *BRCA2* and an HRD score ≥42 (ref. 44), 332 (70%) tumors were predicted responsive to neoadjuvant chemotherapy (Fig. 6a). We also annotated somatic mutations, CNAs and gene fusions using OncoKB[45], and identified 53% of the tumors ($n$ = 246) harboring genetic alterations with known target therapeutic agents at various confidence levels (Supplementary Table 11 and Fig. 6b). It should be noted that none of these molecular

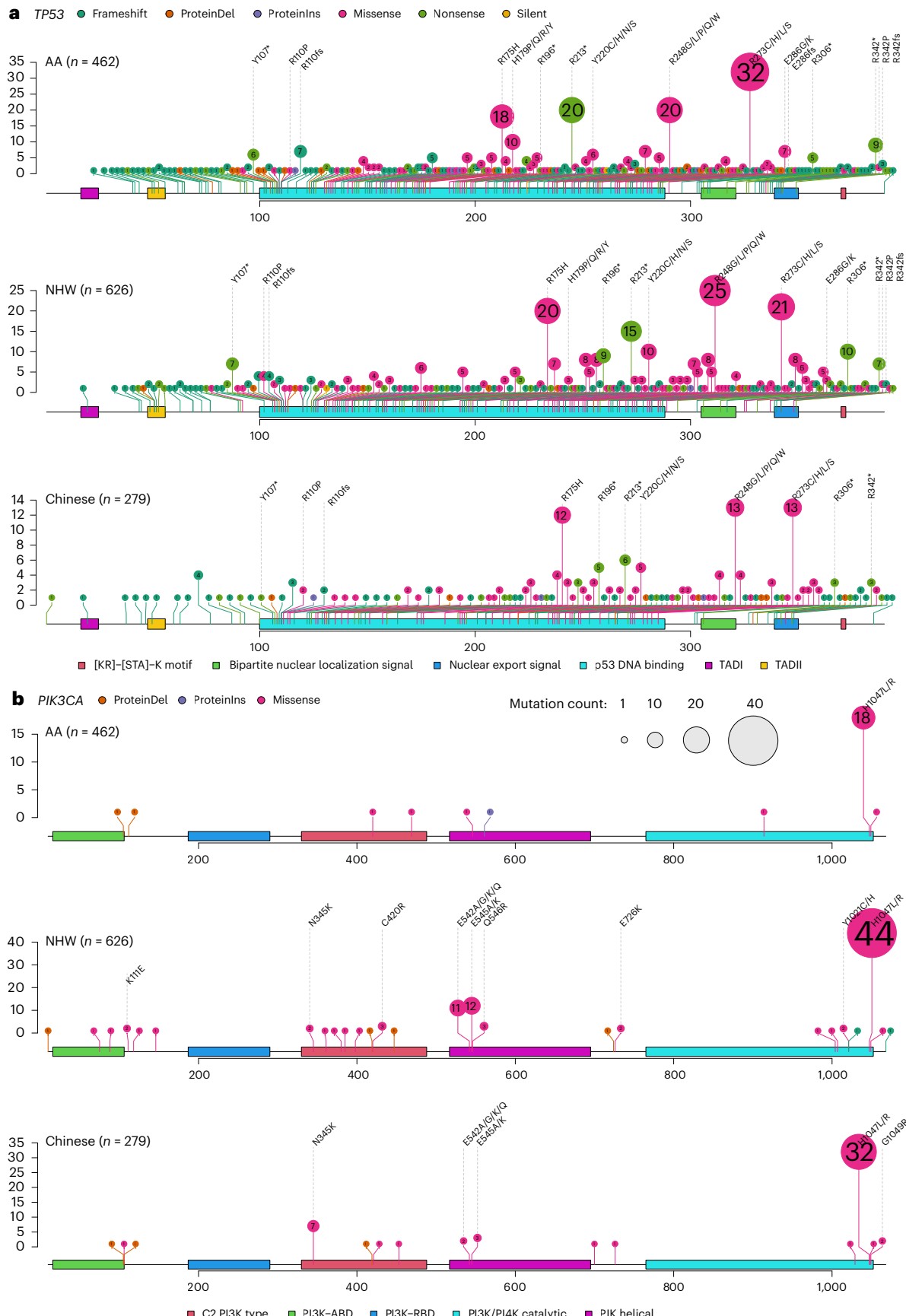

**Fig. 5 | Mutation spectrum of *TP53* and *PIK3CA* in TNBC from AA, Chinese and NHW women. a,b,** Lollipop plot of *TP53* (**a**) and *PIK3CA* (**b**) somatic mutations in TNBC from AA (B-CAUSE, *n* = 462), Chinese (FUSCC, *n* = 279) and NHW women (TCGA, SCAN-B and METABRIC, *n* = 626). ABD, adaptor-binding domain; RBD, RAS-binding domain. The numbers in the circles indicate the number of tumors harboring the mutation in the cohort.

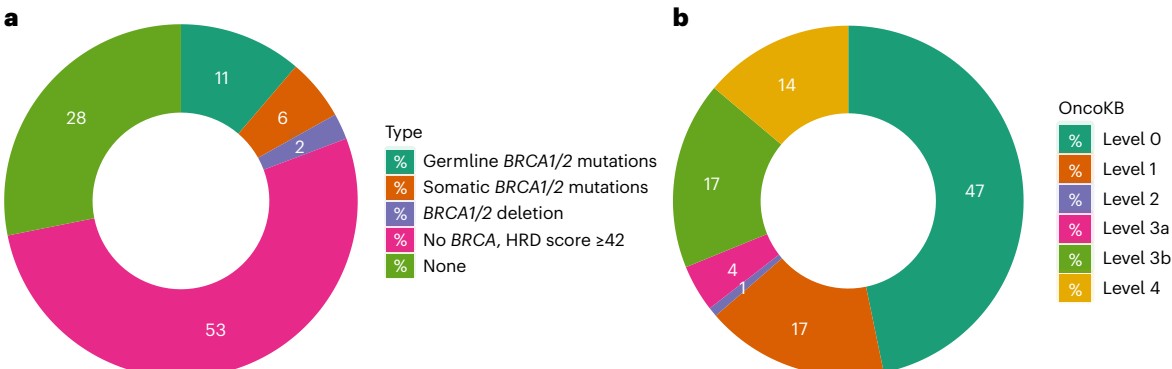

**Fig. 6 | Clinically actionable genomic changes in TNBC from AA women.**
Donut plots of actionable alterations in TNBC from AA women. Numbers in the
plots are percentage of cases classified to each category. **a**, HRD defined
on the basis of germline and somatic mutations in *BRCA1* and *BRCA2*, and
an HRD score ≥42, which predicts response to neoadjuvant chemotherapy.
**b**, Classification of actionable somatic mutations, CNAs or fusion events based
on OncoKB database[45]. Level 1, Food and Drug Administration (FDA)-recognized
biomarker predictive of response to an FDA-approved drug; level 2, standard care
biomarkers predictive of response to an FDA-approved drug; level 3a, compelling
clinical evidence for the biomarker predictive of response to a drug; level 3b,
standard care of investigational biomarker predictive of response to an FDA-
approved or investigational drug; level 4, compelling biological evidence for the
biomarker predictive of response to a drug.

targets nor the associated therapies have been approved for TNBC
treatment. Finally, 163 (35%) and 62 (13%) tumors had copy number gain
and high-level amplification of *CD274* (*PD-L1*), respectively, associated
with higher mRNA expression (Supplementary Fig. 12) and predictive
of response to pembrolizumab[46], an immune checkpoint inhibitor
approved for TNBC treatment.

## Discussion

This large TNBC genomic study based on 462 AA women makes a con-
siderable contribution to the diversification of the cancer genomic
knowledge base. The findings reveal that the mutational landscape of
TNBC in AA women is largely similar to that in Chinese and NHW women
and there was no evidence of associations of mutational features with
African ancestry. Therefore, our results do not support major ancestral
differences in TNBC biology at the level of somatic mutations. Never-
theless, there are several distinctive genomic characteristics noteworthy
in TNBC from AA women.

In comparison to the approximately 80% of mutation frequency in
*TP53* in previous studies[10,11,36], we identified 95% TNBC from AA women
harboring *TP53* mutations. The dominance is more complete than
previously recognized in Chinese and NHW women[10,11,36]. The differ-
ence could be due to TNBC risk factors more common in AA women, or
alternatively, to technical differences in sequencing or variant identi-
fication. Compared with TCGA and other WES-based studies, we used
a larger exome library design with sizable custom contents, longer
150-bp reads and deeper sequencing depth. Variant calling algorithms
could have also contributed to the difference in mutation frequency,
where we integrated four callers followed by a manual review step
for quality control purposes. The fact that we validated essentially
all *TP53* mutations in two orthogonal replication analyses supported
the internal validity of our variant calling. However, caveats should be
taken when comparing our data with those from external studies due
to these technical differences.

We found a partial mutually exclusive pattern between the aging
and HRD mutational signatures in AA women that separated the
tumors into subtypes with distinct demographic, genomic and prog-
nostic features. HRD signature is known to dominate in TNBC[10,11], and
occurred in 53% of the tumors in our cohort that were characterized
by younger age at diagnosis, *BRCA1* and *BRCA2* mutations, higher
mutation burden, more immunogenic and better survival. In contrast,
the aging-related signatures dominated the other half of the tumors,
particularly in older women and those with higher BMI, and were associ-
ated with low mutation burden, less immunogenic and poor survival.

This revelation is rather surprising since TNBC in AA women is often
associated with younger age, and epidemiological evidence for the
association of obesity with TNBC risk has been mixed[47,48]. Our results
suggest that there exist at least two etiological pathways of TNBC in AA
women, one occurring more often in younger patients driven by genetic
predisposition and deficient DNA damage repair, and the other more
often in older patients driven by aging and obesity-related dysregula-
tion. Notably, there was a recent trend of increasing TNBC incidence in
AA women aged ≥55 years in the United States[49]. It is possible that the
higher incidence of TNBC in AA women could be, in part, attributed to
obesity, which is more prevalent in AA women than in white women[50].
It is known that SBS1 is similar to formalin-fixed paraffin-embedded
(FFPE) artifacts after chemical repairment in DNA library preparation[51].
Although we did not perform the repairment step, caution is advised
when interpreting the results related to the SBS1 signature.

The comprehensive genomic characterization of TNBC from AA
women provides an opportunity to assess actionable targets for poten-
tial clinical benefits. Our analysis suggested that 72% of the TNBC from
AA women would be responsive to neoadjuvant chemotherapy. This
estimate is higher than the approximately 60% in white patients[11,52],
which suggests that AA patients should have similar, if not higher,
PCR rate to neoadjuvant chemotherapy than white patients. However,
several earlier studies found the opposite[53,54]. This could be due to
social or structural healthcare barriers faced by AA women[55]. Beyond
chemotherapy, our data also uncovered that 53% of the TNBC tumors in
AA women harbored somatic changes linked with targeted therapeutic
agents. Notably, 12 TNBC patients had *ERBB2* amplification despite
being HER2 negative in clinical assays, which was also found in other
TNBC studies[10,36,37]. This discordance could be due to false-negative
clinical test, false-positive CNV results or tumor heterogeneity between
samples used for clinical assays and those used later for research analy-
sis. It might also reflect the biological heterogeneity of TNBC, as it has
been shown that some TNBC tumors had activating ERBB2 signaling[10].
Further, 48% of the TNBC tumors in AA women had copy number gain
of *CD274*, which predicts better response to pembrolizumab as a part
of the chemoimmunotherapy regimen approved for TNBC. These data
suggest that a majority of the AA patients with TNBC may benefit from
immunotherapy or targeted therapy. A limitation of our analysis was
that all patients were diagnosed and treated before the era of targeted
therapy and immunotherapy for TNBC, and, thus, the therapeutic
prediction was based solely on molecular data. Future clinical trials on
new treatments for TNBC should strive to increase the participation
of AA women[56].

In conclusion, in this large study of TNBC from AA women, we unveiled a high-resolution mutational portrait reminiscent of that in Chinese and NHW women yet with distinctive features, including an almost complete dominance of *TP53* mutations, fewer *PIK3CA* mutations and mutational signature-based subtypes with etiologic and prognostic implications. These findings provide new insights into the disease epidemiology, etiology and therapeutic vulnerability of TNBC from AA women, which also deepens our understanding of this aggressive disease in other populations. Our research highlights the importance of continuous, inclusive scientific efforts to ensure that the rapidly growing cancer knowledge benefits all human populations.

## Online content

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

## Methods

### Patient populations and ethical regulations

B-CAUSE Study pooled data and biospecimens from the following five population-based breast cancer studies with large number of AA women in the United States to investigate epidemiological and clinical significance of tumor somatic mutations. B-CAUSE Study was supported in part by the National Institutes of Health (grant R01 CA228156). For the protection of human participants, all participants provided informed written consent. The study was reviewed and approved by the Institutional Review Board (IRB) at all participating institutions where the patient data and samples came from, specifically Roswell Park Comprehensive Cancer Center (STUDY00000692/BDR 102718), Boston University (H-38636) and Vanderbilt University Medical Center (110190). Detailed information on the participating studies in B-CAUSE is provided in Supplementary Note.

### FFPE tumor tissue processing, DNA/RNA extraction and sample QC

The tumor tissue workflow has been described in detail previously[58]. Dedicated study pathologists reviewed haematoxylin and eosin slides to identify regions enriched with tumors for coring, and the resulting punches were used for DNA and RNA extraction. When tumor blocks were not available, whole sections were macrodissected to trim non-tumor tissues and excess wax where possible before being used for DNA and RNA extraction. For the Women's Circle of Health Study and the Black Women's Health Study, pathological tumor infiltrating lymphocyte score was also recorded by a pathologist based on established guidelines[59]. An optimized DNA/RNA co-extraction protocol on the basis of the Qiagen AllPrep DNA/RNA FFPE kit was used to derive tumor DNA and RNA samples. After extraction, pre-analytic quality control (QC) will be performed using Agilent Bioanalyzer for size distribution check. Matched germline DNA came from those extracted from blood, saliva or mouthwash samples in the parent studies.

### WES

WES were performed by the University of Southern California (USC) Translational Genomics Laboratory using the Keck Genomics Platform and a customized Agilent SureSelect capture probeset to (1) interrogate exons from >20,000 genes based upon the SureSelect Human Whole Exome V6 backbone, (2) to analyze copy number across the genome at 60 kb resolution using additional baits representing Agilent 44k human aCGH probes and (3) to completely tile across known oncogenic breakpoints and tumor suppressor loci to uncover large indels or inversions that can be missed by standard exome sequencing[60]. The total size of the library was 73.499 Mb. Next-generation sequencing (NGS) was conducted using 150-bp paired-end reads on the Illumina NovaSeq 6000 system, with a target sequencing depth of at least 70% bases at 20× for normal DNA and 80% bases at 20× for tumor DNA. A total of 513 pairs of matched tumor and normal DNA samples were sequenced, which generated 0.3 trillion paired-end reads. The median sequencing depth was 154× (range = 14× to 712×), with 89% (range = 16–100%) bases above 20×.

### Bioinformatic processing and variant calling

Bioinformatic processing work was carried out using integrative workflows implemented in the RcwlPipelines package[61]. The bioinformatics analysis began by aligning high-quality paired-end reads passing Illumina RTA filter to the NCBI human reference genome (GRCh37) using BWA-MEM aligner[62]. PCR duplicated reads are marked and removed by using Picard (http://picard.sourceforge.net/). The median map rate was 0.997, median duplication rate was 0.203 and median sequencing depth was 127×. There were 20 tumor samples and 1 normal sample that did not reach the targeted sequencing depth, 10 samples with unmatched tumor-normal sample identity and 4 samples with cryptic relatedness were removed from analysis, leaving 478 tumor-normal pairs. Tumor

purity was estimated using FACETS[63] and 16 samples with low tumor purity (<0.10) were also removed. As a result, 462 tumor-normal pairs were retained in the final analysis. After QC, the median sequencing coverage was 91% (range = 80–100%) for tumor DNA and 87% (70–100%) for normal DNA, respectively, of the bases within the targeted regions with at least 20× depth.

Somatically acquired mutations in tumors were identified from WES data by comparing tumor sequences with individual-matched germline sequences as previously described[64]. Putative SNVs and small indels were initially identified by running four different variant callers, Strelka[65], MuSE[66], VarDict[67] and Mutect[68], through established pipelines in the RcwlPipelines package. All putative SNVs were further filtered based on the following standard set of criteria to remove the common types of false calls: (1) the alternative allele was present in the matching normal sample and the contingency between the tumor and normal samples was not statistically significant, (2) the mutant alleles were only present in one stand and the strand bias was statistically significant, (3) the putative mutation occurred at a site with systematically dropped base quality scores and (4) the reads harboring the mutant allele were associated with poor mapping quality. Ambiguous cases were manually inspected to ensure accuracy using Bambino viewer[69]. Putative indels were evaluated by a re-alignment process to filter out potential false calls introduced by unapparent germline events, mapping artifacts and homopolymer. The identified somatic mutations were compared with the public human germline databases, including dbSNP[70], 1000 Genomes Project[71], National Heart, Lung and Blood Institute's Exome Sequencing Project[26] and the Genome Aggregation Database (gnomAD)[72]. The identified mutations were annotated by ANNOVAR[73] using the Ensembl database[74]. Multiple nucleotide variants were identified and annotated using MAC[75]. Tumor mutation burden was calculated as the number of SNVs per Mb of genomic region sequenced. Hypermutated samples were identified as those with a mutation rate ≥10 SNVs per Mb[13].

### Confirmation of TP53 mutations

Two orthogonal approaches were used to confirm TP53 mutations identified by WES data. First, for the 260 tumor samples with available matched transcriptome data, the mutant and wild-type (WT) reads were extracted from the transcriptome data to determine the expression level of the mutant allele. Second, an independent resequencing assay by TAS that covered the entire coding region of TP53 was carried out using 338 tumor samples and eight normal samples with adequate amount of DNA samples left after WES. Each putative TP53 mutation identified from WES was revisited in the TAS data to extract the mutant and WT read counts from the BAM file as previously described[76]. A mutation was considered as confirmed if its variant allele fractions in the tumor TAS were substantially higher than in the control sample using a highly stringent criterion ($P < 10 \times 10^{-10}$, Fisher's exact test). For the samples with no TP53 mutation detected by the original WES, de novo mutation detection was performed in the TAS data by comparing each tumor with a pool of all normal samples as previously described[77], which identified two new TP53 mutations, including one SNV and one indel. These two newly identified TP53 mutations were manually inspected using Bambino viewer[69] and the indel was further confirmed using IndelPost[78].

### Somatic copy number alteration analysis

Tumor segmentation and somatic copy number alteration (SCNA) calling were carried out using FACETS[63] based on the WES data from a customized sequencing library originally designed with additional baits representing Agilent 44k human aCGH array probes. GISTIC2 (ref. 79) was then used to identify substantially amplified or deleted SCNAs with a residual false discovery rate (FDR) $q < 0.25$. Genes residing within the wide peak regions were identified and a GISTIC threshold CN value (−2, −1, 0, 1, 2) indicating the amplitude (high-level deletion,

low-level deletion, diploid, low-level amplification and high-level amplification) of the CNA was assigned to each tumor.

## SMG analysis

MutSigCV[17], MutSig2CV[18] and MuSiC[19] were used to identify genes mutated at a substantially higher rate than the background mutation rate, with significance set at FDR $q$ < 0.20. The final list of SMGs were determined as those that were substantial in two or more of the three programs used. Known TNBC genes are defined based on SMGs identified in prior studies focusing specifically on TNBC; known breast cancer genes are defined based on SMGs identified in prior studies focusing on breast cancer but not specifically on TNBC; and known pan-cancer genes are defined based on SMGs in prior pan-cancer studies.

## Pathogenic germline mutation analysis

Germline WES data from AA TNBC patients in B-CAUSE were used for analysis of germline mutations in nine known breast cancer genes, including *ATM*, *BRCA1*, *BRCA2*, *CDH1*, *CHEK2*, *NF1*, *PALB2*, *PTEN* and *TP53*. Pathogenicity data were extracted from ClinVar[28]. All germline variants identified were queried in public population reference databases, including gnomAD[25], TOPMed[26], National Institutes of Health All of US[24] and the Regeneron Genetics Center. Whenever possible, reference data from populations of African ancestry were used for comparison. To formally test the allele frequency of the germline variants in TNBC patients in B-CAUSE against the reference datasets, chi-square or Fisher's exact test was used, whichever was appropriate. Finally, the germline variants were cross-referenced with those characterized in functional studies, including *BRCA1* (ref. 29), *BRCA2* (ref. 80), *CHEK2* (ref. 81) and *TP53* (ref. 82).

## GSEA

To identify common pathways somatically altered in TNBC, GSEA was carried out based on 218 driver genes harboring recurrent mutations ≥1% in the cohort using Hallmark gene sets from the MSigDB[83]. An FDR $q \leq 0.05$ was considered substantial.

## Mutational signature analysis

Mutational signature analysis was performed with sigProfiler-Extractor[84], which allows de novo extraction of operative mutational signatures and subsequently decomposition to known COSMIC mutational signatures[30]. SBS and indel signatures were extracted, whereas double base substation mutations were sparse in the WES data and not analyzed. The five tumor samples with hypermutation phenotype were excluded from mutational signature analysis. To define SBS-based TNBC subtypes, the combined activity of the aging-related signatures SBS1 and SBS5, along with the HRD-related signature SBS3, was categorized into tertiles and assessed for association with patient survival outcomes. Based on the survival differences, the aging signature and the HRD signature were further collapsed into two groups (T1 and T2 versus T3). The resultant binary classifications of the two signatures were used to create a combined SBS-based TNBC subtype between aging and HRD signatures. SBS subtype 1 had low aging and high HRD signatures, subtype 2 had low aging and low HRD signatures, subtype 3 had high aging and low HRD signatures and subtype 4 had high aging and high HRD signatures.

## Genetic ancestry

Global genetic ancestry was estimated based on germline genotype data derived from WES of the matched normal samples using ALStructure[85]. Data of reference ancestral populations were obtained from the 1000 Genomes Project[71] and used as benchmarks in the estimation.

## RNA sequencing and bioinformatic processing

RNA sequencing for tumors from the Women's Circle of Health Study and the Black Women's Health Study was performed using the Agilent SureSelectXT Exome RNA kit, which is optimized for library preparation from degraded RNA samples in FFPE tissues. NGS was then conducted on an Illumina NovaSeq 6000 sequencer, yielding an average of 50 million 150-bp paired-end reads per sample. Sequencing work for tumors from Vanderbilt studies was performed using the BGI-Seq platform with 100-bp read length. Raw reads were demultiplexed and preprocessed by using FastQC[86] and Cutadapt[87]. The remaining reads were mapped to the GRCh38 and reference transcriptome GENCODE using STAR-Aligner[88]. Additional QC steps were performed to identify library preparation problems using RSeQC[89] and/or RNA degradation issues using mRIN[90]. RSEM[91] was used to quantify gene expression as transcript per million reads and fragments per kilobase per million mapped reads. Batch correction was conducted using ComBat-seq[92] and wherever appropriate, study site was adjusted as a covariate in multivariable models. For sample-level and gene-level filtering, we excluded genes with low or no expression; those with either an interquartile range equal to zero or a sum across samples ≤1; and outliers based on the principal component analysis and the RNA degradation threshold.

## RNA fusion events

RNA fusions were characterized in 260 TNBC samples with transcriptomic data available. Arriba[93] and STAR-Fusion[94] programs were both used. For stringent filtering, high-confidence chimeric transcripts identified from each program were intersected and only those overlapping between the two programs were reported. Genomic plots of the fusion mutations and resultant protein domains are generated using script provided in Arriba.

## Neoantigen analysis

Neoantigens are aberrant peptides introduced by nonsynonymous somatic mutations in tumors, some of which can be recognized by the immune system and illicit antigen-specific immune response in the tumor microenvironment and thus are predictive of cancer immunotherapy. To characterize neoantigen profiles in TNBC tumors, we cross-referenced the predicted binding affinity ($IC_{50}$ < 500 nM) of 8-mer to 15-mer amino acid sequences generated by each nonsynonymous missense mutation with mRNA expression (read count ≥ 1) to estimate the number of aberrant peptides as potential neoantigens.

## Potential molecular alterations

Three approaches were applied to identify actionable alterations in TNBC. First, to predict response to neoadjuvant chemotherapy, an HRD score was calculated as the sum of the three factors, including loss of heterozygosity, telomeric allelic imbalance and large-scale state transition scores, using BAM files of tumor samples by the scarHRD R package with default parameters[57]. An established cutoff value of HRD score ≥42 (ref. 44) was used to classify tumor with HRD-high status. Further, patients carrying either pathogenic germline variants or somatic mutations in *BRCA1* and *BRCA2* were automatically classified as HRD-high status. Second, all somatic mutations, SCNAs and RNA fusion events were annotated for targeted therapeutic agents using the OncoKB[45,95], which classified the molecular alterations into four levels based on the strength of the evidence. Third, copy number gain of *CD274* encoding for PD-L1 was analyzed as predictive marker for response to immune checkpoint inhibitors.

## Publicly available TNBC mutation datasets

For comparison to somatic mutations in TNBC patients from AA women in B-CAUSE, publicly available data were aggregated from Asian women ($n$ = 279) in FUSCC and NHW women in TCGA ($n$ = 69), the SCAN-B ($n$ = 254) and the METABRIC ($n$ = 320). More details of the public datasets are provided in Supplementary Note.

## Statistical analysis

This study was designed as an observational analysis to characterize the tumor somatic mutational landscape of TNBC in AA women, and thus is

not considered a randomized study. To maximize statistical power, all AA women with TNBC from the participating studies with available tumor and matched normal DNA samples were included, and thus no statistical method was used to predetermine sample size. Except for samples removed due to poor data quality in the quality control steps, no other samples were excluded. Because all samples were from TNBC cases and tumor and normal samples were sequenced separately to reach different targeted sequencing depth, the laboratory performing the sequencing was not blinded to tumor versus normal status of the samples.

Descriptive statistics were computed as mean and s.d. for numeric variables and count and percentage for categorical variables. Correlation analyses were performed using the Pearson method. For comparisons of means or medians between groups, $t$ tests, analysis of variance or equivalent nonparametric version tests were applied. For survival analysis, Kaplan–Meier curves were plotted with $P$ values derived from log-rank tests, followed by Cox proportional hazard models with adjustment for covariates (age at diagnosis, study and cancer stage). All analyses were performed in R (v.4.4.0) and a two-side $P \leq 0.05$ was considered statistically significant, unless otherwise specified.

### Reporting summary
Further information on research design is available in the Nature Portfolio Reporting Summary linked to this article.

## Data availability
TCGA data: https://xena.ucsc.edu/public/. The SCAN-B: https://data.mendeley.com/datasets/2mn4ctdpxp/3. The METABRIC: https://github.com/cBioPortal/datahub/tree/master/reference_data/gene_panels. The FUSCC: https://figshare.com/articles/dataset/A_comprehensive_genomic_and_transcriptomic_dataset_of_triple-negative_breast_cancers/19783498/5. Human Reference Genome (GRCh37): https://www.ncbi.nlm.nih.gov/datasets/genome/GCF_000001405.13/

The WES data of matched tumor and normal TNBC samples in B-CAUSE have been deposited to the database of Genotypes and Phenotypes (dbGaP), with access number phs003962.v1.p1.

## Code availability
No original or custom code or special software or program was used that was central to the data analyses. The following standard open-source software and programs were used in the data analysis: RcwlPipelines R package (v.1.20.0), BWA-MEM aligner (v.0.7.18), Strelka2 (v.2.9.10), MuSE (v.2.0), VarDict (v.1.1), Mutect2 (v.4.1.0.0), FACETS (v.0.6.2), GISTIC2 (v.2.0.23), MutSigCV (v.1.3.01), MutSig2CV, MuSiC (v.0.4), sigProfilerExtractor (v.1.1.24), ALStructure R package, FastQC (v.0.12.1), Cutadapt (v.2.0), STAR-Aligner (v.2.7.11b), RSeQC (v.5.0.1), mRIN (v.1.2.0), RSEM (v.1.3.3), sva-ComBat-seq R package (v.3.52.0), Arriba (v.2.4.0), STAR-Fusion (v.1.13.0), scarHRD R package (v.0.1.1), GSEA (v.4.3.3), MSigDB (v.2023.2), OncoKB API (v.3.4.1) and R programming (v.4.4.0).

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

## Acknowledgements

This research was supported in part by the National Institutes of Health (grants R01 CA228156 to S.Y., J.R.P., W.Z. and J.D.C.; R01 CA100598 to C.B.A.; R01 CA185623 to C.C.H. and E.V.B.; R01 CA164974 to J.R.P.; UM1 CA164974 to J.R.P.; R01 CA092447 to W.Z.; U01 CA202979 to W.Z.; R01 CA100374 to W.Z. and U54 CA91405 to W.Z.) and Komen Foundation (SAC220228 to J.R.P.). Sample preparation, RNA sequencing, targeted sequencing assays and bioinformatic analysis at Roswell Park Comprehensive Cancer Center were conducted at the Biobanking and Laboratory Service, Genomics Shared Resource and Bioinformatics Shared Resource, which are supported in part by the Roswell Park Cancer Center Support Grant (P30 CA016056 to C.S. Johnson). Data collection and sample preparation work for Vanderbilt samples were performed by the Survey and Biospecimen Shared Resource, supported in part by the Vanderbilt-Ingram Cancer Center (P30 CA068485 to B.H. Park). High-performance computing data analyses were conducted using the Center for Computational Research at the University at Buffalo. C.B.A. received support from the Breast Cancer Research Foundation. J.R.P. and M.E.B. received support from the Karin Grunebaum Cancer Research Foundation. The content is solely the responsibility of the authors and does not necessarily represent the official views of the funding agents. The funders had no role in study design, data collection and analysis, decision to publish or preparation of the manuscript.

## Author contributions

S.Y., S.L., C.B.A., J.R.P., W.Z. and J.D.C. conceived of and designed the study. M.S., C.C.H., E.V.B., C.B.A., J.R.P. and W.Z. recruited the study participants and collected the data and specimens. S.Y., L.W., Q.H., D.W.C., Z.M., G.R.Z., J.P., Q.C., J.L., M.E.B., Y.J., W.D., M.M., J.C., R.P.O. and T.K. managed sample and data preparation, and carried out quality control. S.Y., L.W., Q.H., Z.M., P.N.F., M.L., M.M., J.W. and Q.Z. analyzed the data. S.Y., L.W., Q.H., Z.M., T.K., S.G., K.T., N.K., S.L., C.B.A., J.R.P., W.Z. and J.D.C. interpreted the findings. S.Y., L.W., P.N.F., S.L., C.B.A., J.R.P., W.Z. and J.D.C. drafted or substantively revised the paper. S.Y., J.R.P., W.Z. and J.D.C. supervised the project.

## Competing interests

The authors declare no competing interests.

## Additional information

**Extended data** is available for this paper at https://doi.org/10.1038/s41588-025-02322-y.

**Correspondence and requests for materials** should be addressed to Song Yao.

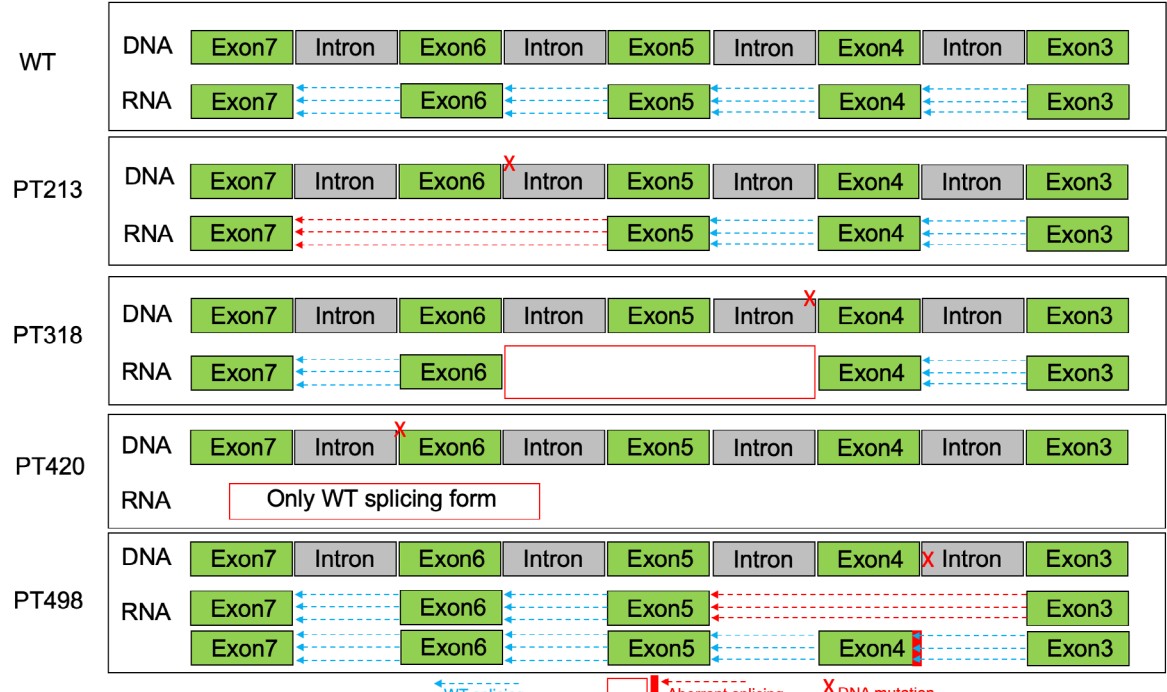

**Extended Data Fig. 1 | Impact of *TP53* intronic mutations on RNA transcript splicing.** Four triple-negative breast cancer (TNBC) samples carried only an intronic mutation in *TP53* with matched tumor transcriptomic data available. To explore the functional impact of the mutations, the splicing patterns of the transcripts of these samples are examined, with one sample harboring no *TP53* mutation as the wild-type (WT) reference. DNA mutations are indicated by a red 'X' mark on DNA sequence at the top, and wild-type (WT) splicing patterns indicated by blue color and aberrant splicing patterns indicated by red color on the RNA sequence at the bottom.

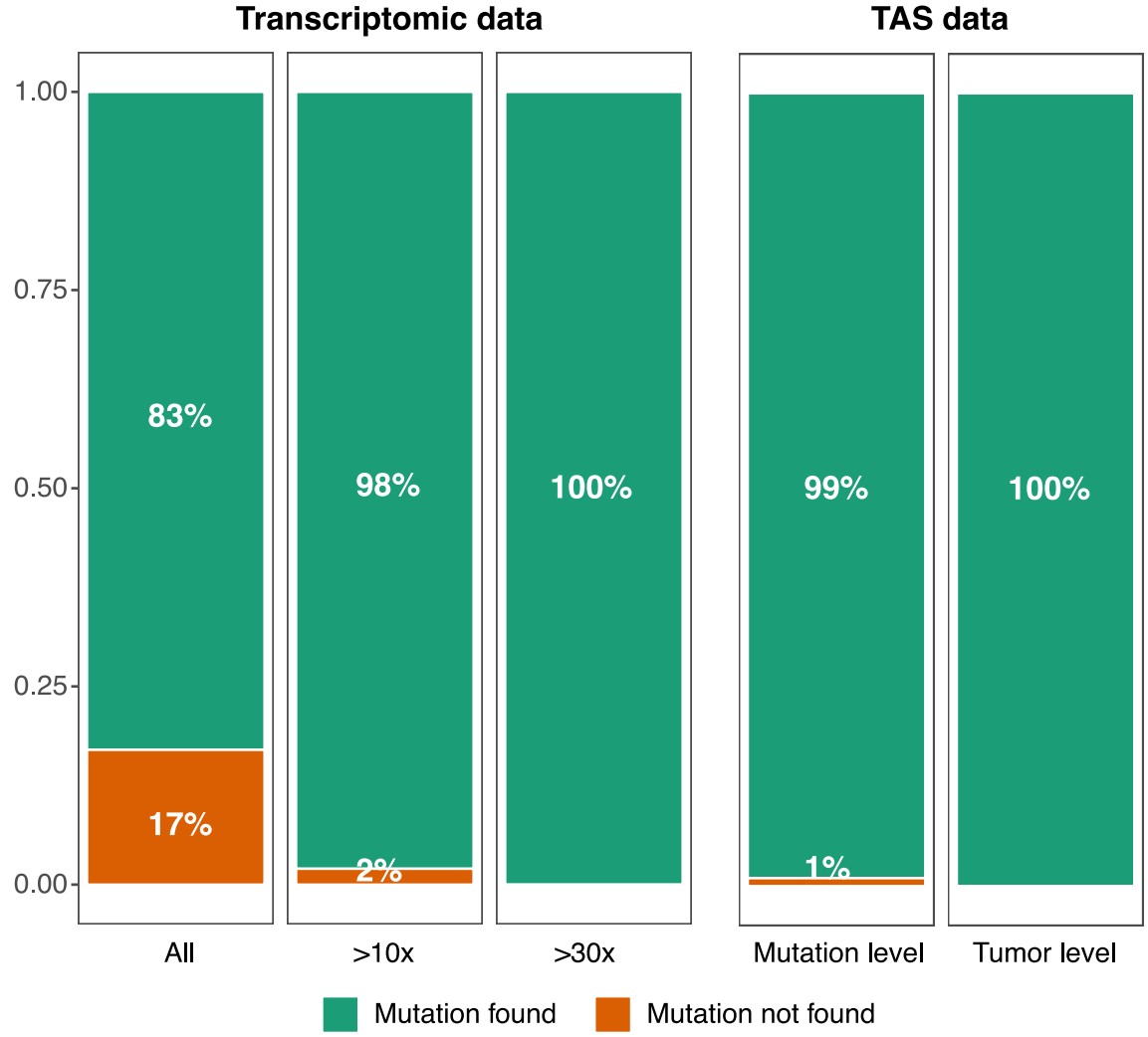

**Extended Data Fig. 2 | Confirmation of TP53 mutations in transcriptomic data and ultra-deep sequencing.** Two orthogonal approaches were used to validate *TP53* mutations identified from whole-exome sequencing analysis—first, using tumor transcriptomic data available from 260 matched RNA samples, and second, using targeted amplicon sequencing (TAS) performed on 338 tumors with DNA available. The numbers shown in the bar indicate the percentage of samples with mutations replicated (green) or not replicated (orange). For transcriptomic data, the results are shown for all samples and by the sequencing depth of TP53 region. For TAS data, the results are shown at both mutation level and tumor level.

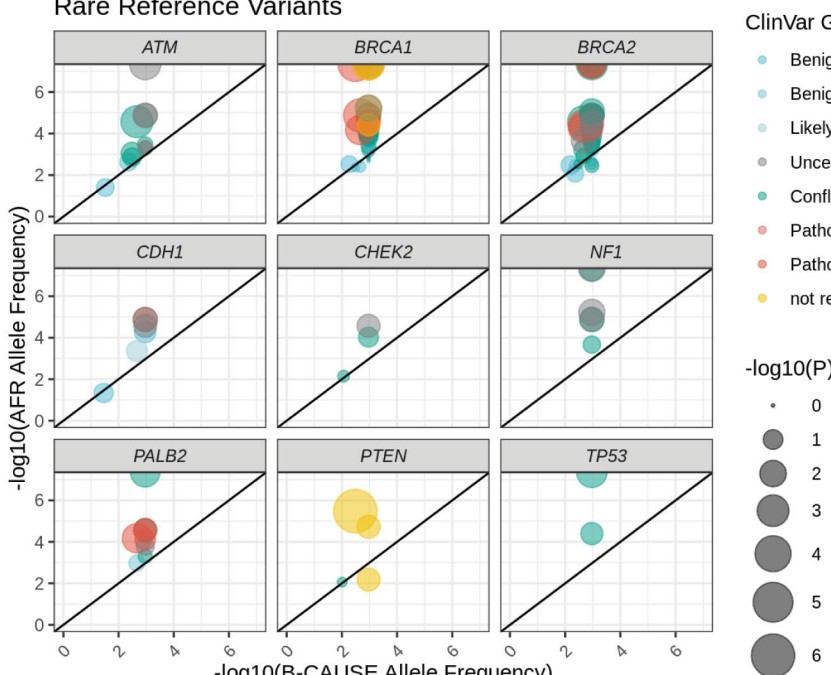

**Extended Data Fig. 3 | Minor allele frequency of germline mutations in known breast cancer genes in B-CAUSE TNBC patients and reference populations.** The minor allele frequency in B-CAUSE data (x-axis) and AFR reference data (y-axis) are log-transformed and negated and germline variants are color coded according to ClinVar classification. P-values from two-sided Fisher's exact test are indicated by the dot diameter.

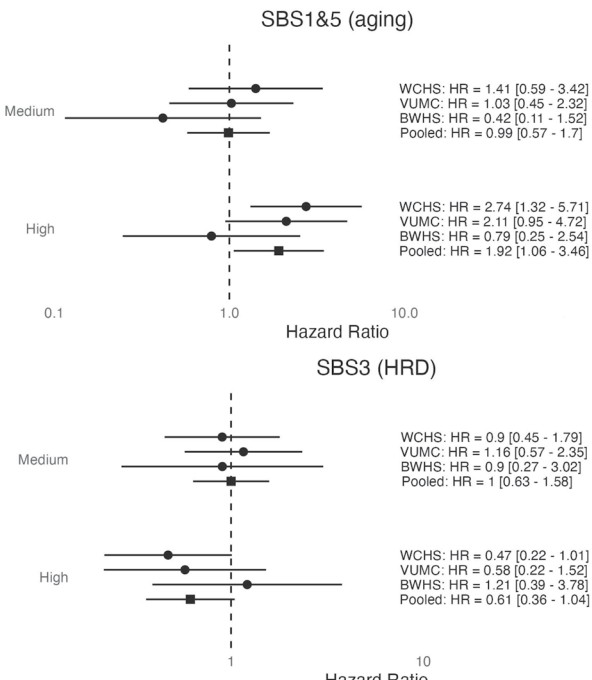

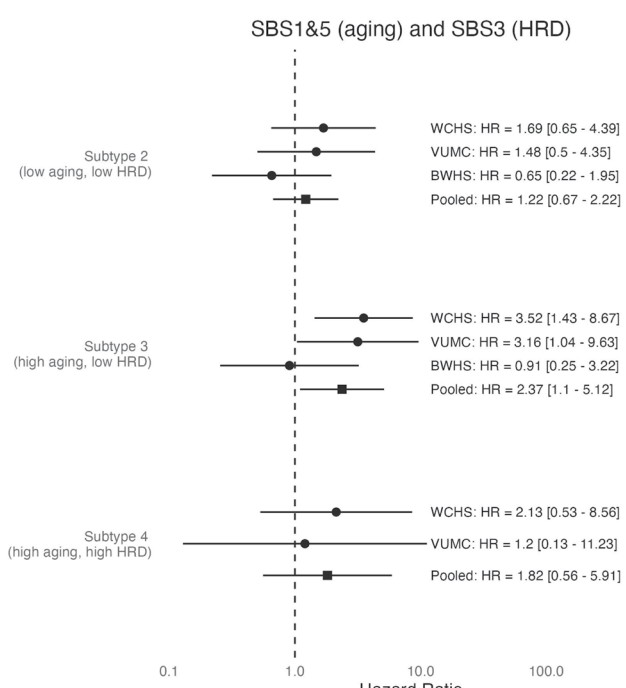

**Extended Data Fig. 4 | Forest plots of overall survival with mutational signatures and SBS-based subtypes.** The low, medium and high levels of SBS1&5 (aging) and SBS3 (HRD) were determined by tertiles of the signatures. The low levels were used as the reference group (not shown in the figures). In the analysis of the combined SBS1&5 and SBS3 group, the subtype 1 (low aging and high HRD) was used as the reference group (not shown in the figure). For subtype 4, the model did not converge in BWHS and the results are not shown. The dots indicate the point estimates of hazards ratio (HR) with the bars indicate the lower and upper 95% confidence intervals (CI). The squares are pooled HR estimates from meta-analyses across studies. Sample size for each study was: WCHS: n = 146; VUMC: n = 84; BWHS: n = 72.

# Reporting Summary

## Statistics

For all statistical analyses, confirm that the following items are present in the figure legend, table legend, main text, or Methods section.

| n/a | Confirmed | |
|---|---|---|
| ☐ | ☒ | The exact sample size (*n*) for each experimental group/condition, given as a discrete number and unit of measurement |
| ☐ | ☒ | A statement on whether measurements were taken from distinct samples or whether the same sample was measured repeatedly |
| ☐ | ☒ | The statistical test(s) used AND whether they are one- or two-sided<br>*Only common tests should be described solely by name; describe more complex techniques in the Methods section.* |
| ☐ | ☒ | A description of all covariates tested |
| ☐ | ☒ | A description of any assumptions or corrections, such as tests of normality and adjustment for multiple comparisons |
| ☐ | ☒ | A full description of the statistical parameters including central tendency (e.g. means) or other basic estimates (e.g. regression coefficient) AND variation (e.g. standard deviation) or associated estimates of uncertainty (e.g. confidence intervals) |
| ☐ | ☒ | For null hypothesis testing, the test statistic (e.g. *F*, *t*, *r*) with confidence intervals, effect sizes, degrees of freedom and *P* value noted<br>*Give P values as exact values whenever suitable.* |
| ☒ | ☐ | For Bayesian analysis, information on the choice of priors and Markov chain Monte Carlo settings |
| ☒ | ☐ | For hierarchical and complex designs, identification of the appropriate level for tests and full reporting of outcomes |
| ☐ | ☒ | Estimates of effect sizes (e.g. Cohen's *d*, Pearson's *r*), indicating how they were calculated |

*Our web collection on statistics for biologists contains articles on many of the points above.*

## Software and code

Policy information about availability of computer code

| Data collection | No specific code or software was used as a central part of the data collection for this study. |
|---|---|
| Data analysis | No custom code or special software/program was used that was central to the data analyses. The following standard open-source software and programs were used in the data analysis:<br><br>RcwlPipelines R package (v.1.20.0)<br>BWA-MEM aligner (v.0.7.18)<br>Strelka2 (v.2.9.10)<br>MuSE (v.2.0)<br>VarDict (v.1.1)<br>Mutect2 (v.4.1.0.0)<br>FACETS (v.0.6.2)<br>GISTIC2 (v.2.0.23)<br>MutSigCV (v.1.3.01)<br>MutSig2CV<br>MuSiC (v.0.4)<br>sigProfilerExtractor (v.1.1.24)<br>ALStructure R package<br>FastQC (v.0.12.1)<br>Cutadapt (v.2.0)<br>STAR-Aligner (v.2.7.11b) |

RSeQC (v.5.0.1)
mRIN (v.1.2.0)
RSEM (v.1.3.3)
sva-ComBat-seq R package (v.3.52.0)
Arriba (v.2.4.0)
STAR-fusion (v.1.13.0)
scarHRD R package (v.0.1.1)
GSEA (v.4.3.3)
MSigDB (v.2023.2)
OncoKB API (v.3.4.1)
R programming (v.4.4.0)

For manuscripts utilizing custom algorithms or software that are central to the research but not yet described in published literature, software must be made available to editors and reviewers. We strongly encourage code deposition in a community repository (e.g. GitHub). See the Nature Portfolio guidelines for submitting code & software for further information.

# Data

Policy information about availability of data

All manuscripts must include a data availability statement. This statement should provide the following information, where applicable:

- Accession codes, unique identifiers, or web links for publicly available datasets
- A description of any restrictions on data availability
- For clinical datasets or third party data, please ensure that the statement adheres to our policy

The Cancer Genome Atlas (TCGA) data: https://xena.ucsc.edu/public/. The Sweden Cancerome Analysis Network – Breast (SCAN-B): https://data.mendeley.com/datasets/2mn4ctdpxp/3. The Molecular Taxonomy of Breast Cancer International Consortium (METABRIC): https://github.com/cBioPortal/datahub/tree/master/reference_data/gene_panels. The Fudan University Shanghai Cancer Center (FUSCC): https://figshare.com/articles/dataset/A_comprehensive_genomic_and_transcriptomic_dataset_of_triple-negative_breast_cancers/19783498/5. Human Reference Genome (GRCh37): https://www.ncbi.nlm.nih.gov/datasets/genome/GCF_000001405.13/
The whole-exome sequencing data of matched tumor and normal TNBC samples in B-CAUSE has been deposited to the database of Genotypes and Phenotypes (dbGaP), with access number phs003962.v1.p1.

# Human research participants

Policy information about studies involving human research participants and Sex and Gender in Research.

| Reporting on sex and gender | All patients were self-identified as African American females, because this study focuses specifically on breast cancer in African American females due to their disproportionately high disease burden. Breast cancer in biological males is rare. |
|---|---|
| Population characteristics | The patient descriptive characteristics are summarized in Extended Data Table 1. All patients were self-identified as Black females. The average (±sd) age at diagnosis was 53 (±11) years, with 38% before age 50. Twenty-two percent (22%) reported first-degree family history of breast cancer and 4% reported family history of ovarian cancer. Most cases were diagnosed at stage I (40%) or II (44%) and poorly differentiated (90%). |
| Recruitment | This study did not conduct any new patient recruitment work, but pooled existing data and biospecimens from the following five population-based breast cancer studies with large number of African American (AA) females in the US to investigate epidemiological and clinical significance of tumor somatic mutations.<br><br>The Women's Circle of Health Study (WCHS) is a case-control study initiated in 2002 to examine risk factors for aggressive breast cancer in AA and White females. Cases were first identified from hospitals in metropolitan New York City and subsequently through New Jersey State Cancer Registry using rapid case ascertainment. Upon consent, patients complete an in-depth interview on known and suspected risk factors for breast cancer. Follow-up for mortality outcomes was conducted by data linkage with the New Jersey State Cancer Registry as part of the Women's Circle of Health Follow-up Study (WCHSFS).<br><br>The Black Women's Health Study (BWHS) is a U.S.-based prospective cohort study that began in 1995 when 59,000 self-identified AA females 21-69 years of age completed a baseline health questionnaire59. Updated information on breast cancer risk factors and self-report of new breast cancers are obtained via biennial follow-up questionnaires. Incident breast cancers are also ascertained through linkage to 24 state cancer registries that, together, cover the state of residence for >95% of participants. Medical record and cancer registry data are sought for all participants who report a diagnosis of breast cancer.<br><br>The Southern Community Cohort Study (SCCS) was initiated in 2002 to study health disparities and enrolled approximately 86,000 adults in 12 southeastern states60. Nearly 70% of participants are AA. Extensive epidemiological data at baseline were collected. Incident breast cancer cases are identified via linkage to state cancer registries, and clinical data from the cancer registries and supplemented by pathology reports and medical records.<br><br>The Nashville Breast Health Study (NBHS)61 is a population-based case-control study of incident breast cancer among females in the Nashville area initiated in 2004 and later expanded to the entire state to increase the sample size for AA females. Eligible cases were identified primarily through the Tennessee State Cancer Registry. All participants were interviewed to obtain information related to risk factors for breast cancer. |

The Southern Tri-State Breast Health Study (STSBHS) was launched in 2013 to recruit AA breast cancer patients in Tennessee, Georgia, and South Carolina using the established protocols and study instruments developed in the NBHS.

| Ethics oversight | The study protocol for human subject protection was reviewed and approved by the Institutional Review Boards of all participating institutions. The IRB protocol numbers are: Roswell Park (STUDY00000692/BDR 102718); Boston University (H-38636); Vanderbilt University Medical Center (#110190). |

Note that full information on the approval of the study protocol must also be provided in the manuscript.

# Field-specific reporting

Please select the one below that is the best fit for your research. If you are not sure, read the appropriate sections before making your selection.

☒ Life sciences  ☐ Behavioural & social sciences  ☐ Ecological, evolutionary & environmental sciences

For a reference copy of the document with all sections, see nature.com/documents/nr-reporting-summary-flat.pdf

# Life sciences study design

All studies must disclose on these points even when the disclosure is negative.

| Sample size | We assembled 512 triple-negative breast cancer with existing paired tumor and normal samples available from self-identified African American females for whole-exome sequencing from five population-based breast cancer studies in the US. These were all the eligible cases from these studies. No prior sample size calculation was performed. Of those patients, 260 patients had tumor transcriptomic data from RNA sequencing. In addition, we also included TNBC cases with tumor sequencing data from publicly available data sources, including 279 Asian cases from Fudan University Shanghai Cancer Center (FUSCC), 254 non-Hispanic White cases from the Sweden Cancerome Analysis Network – Breast (SCAN-B), 69 non-Hispanic White cases from The Cancer Genome Atlas (TCGA), and 320 non-Hispanic White cases from the Molecular Taxonomy of Breast Cancer International Consortium (METABRIC). These are the all the cases publicly available for comparison with African American cases in our data. No statistical methods were used to predetermine the sample size. |
| Data exclusions | During the data processing and QC processes for the whole-exome sequencing data, there were 20 tumor samples and one normal sample that did not reach the targeted sequencing depth, 10 samples with unmatched tumor-normal sample identity, and four samples with cryptic relatedness were removed from analysis, leaving 478 tumor-normal pairs. Tumor purity was estimated using FACETS and 16 samples with low tumor purity (<0.10) were also removed. As a result, 462 tumor-normal pairs were retained in the final analysis. |
| Replication | Two orthogonal methods were used to confirm TP53 mutations, one using transcriptomic data available from 260 cases, and the other using targeted amplicon sequencing in 338 cases where adequate tumor DNA was available after whole-exome sequencing. Both validation analyses were successful to confirm the TP53 mutations. For analyses of mutational signature-based subtypes with patient survival, meta-analysis across the three studies were performed, which showed consistent results across studies. |
| Randomization | This is an observational study to characterize the mutational landscape of TNBC in African American women, and randomization is not relevant. The analyses were adjusted for covariates to minimize possible confounding effects. |
| Blinding | Laboratory technicians who performed the sample processing and sequencing work were blinded with no information disclosed that could potentially bias their work. |

# Reporting for specific materials, systems and methods

We require information from authors about some types of materials, experimental systems and methods used in many studies. Here, indicate whether each material, system or method listed is relevant to your study. If you are not sure if a list item applies to your research, read the appropriate section before selecting a response.

## Materials & experimental systems

| n/a | Involved in the study |
|-----|----------------------|
| ☒ ☐ | Antibodies |
| ☒ ☐ | Eukaryotic cell lines |
| ☒ ☐ | Palaeontology and archaeology |
| ☒ ☐ | Animals and other organisms |
| ☒ ☐ | Clinical data |
| ☒ ☐ | Dual use research of concern |

## Methods

| n/a | Involved in the study |
|-----|----------------------|
| ☒ ☐ | ChIP-seq |
| ☒ ☐ | Flow cytometry |
| ☒ ☐ | MRI-based neuroimaging |

