## [Peer Review File · Nature Genetics]

Mutational Landscape of Triple-Negative Breast Cancer in African American Women

Corresponding Author: Dr Song Yao

Version 0:

Decision Letter:

12th Mar 2025

Dear Dr Yao,

Your Article, "Mutational Landscape of Triple-Negative Breast Cancer in US Black Women" has now been seen by 2 referees. You will see from their comments below that while they find your work of interest, some important points are raised. We are interested in the possibility of publishing your study in Nature Genetics, but would like to consider your response to these concerns in the form of a revised manuscript before we make a final decision on publication.

We therefore invite you to revise your manuscript taking into account all reviewer and editor comments. Please highlight all changes in the manuscript text file using the tracked-changes function. At this stage we will need you to upload a copy of the manuscript in MS Word .docx or similar editable format.

*2) If you have not done so already please begin to revise your manuscript so that it conforms to our Article format instructions, available

[here](http://www.nature.com/ng/authors/article_types/index.html).

*3) Include a revised version of any required Reporting Summary: <https://www.nature.com/documents/nr-reporting-summary.pdf>

EXTENDED DATA FIGURES

Link Redacted

We hope to receive your revised manuscript within four to eight weeks. If you cannot send it within this time, please let us know.

Sincerely,

Safia Danovi, PhD
Senior Editor, Nature Genetics
ORCID: 0009-0007-7822-5479

Referee expertise:

Referee #1: cancer genomics, multi-ancestry datasets

Referee #2: TNBC genomics

Reviewers' Comments:

Reviewer #1 (Remarks to the Author):

Yao et al conducted an important study of whole-exome sequencing on a large cohort of triple-negative breast cancer (TNBC) patients of African ancestry. This effort addresses the critical need to understand the genetic basis of a disease that disproportionately affects Black women. The paper is well-written, and the data analysis is comprehensive. The most interesting finding is that TP53 is mutated in 95% of TNBC with African ancestry. There are several issues should be addressed. My specific comments are below.

1. TERT coding mutations have not been demonstrated as cancer drivers in large-scale genomic studies. Do these mutations functionally impact TERT expression? Are there any hotspot mutations or are those mutations clustered in a functional domain, suggesting they may act as driver mutations? Additionally, why do these mutations appear significant only in MuSiC but not in MutSigCV? It may be worth running MutSig2CV, an advanced version of MutSigCV that accounts for additional confounding factors to resolve this discrepancy and rule out the possibility of passenger events.
2. The study cohort is substantially larger than the TCGA triple-negative cohort of non-Hispanic white samples, which may lead to confounding results due to the lack of appropriate controls. For instance, mutations in GNAS and GATA3 are well-established breast cancer genes found across all populations. However, the statement in lines 324–326 could mislead readers into thinking these genes are exclusive to Black women.
3. Similarly, TP53 mutation analysis is limited to comparing their cohort with TCGA, while including other existing cohorts (i.e. AACR GENIE) can add more controls to increase the power.
4. The survival analyses pool patients from multiple institutions and hospitals and analyze them together. Differences in neighborhood demographics and access to quality breast cancer care could confound the survival results. These disparities should be accounted for to ensure accurate conclusions. For example, survival analysis at different site should be analyzed separately, or meta-analyzed.
5. The criteria for defining pathogenic germline variants specifically for African ancestry patients are unclear. Given the lack of germline sequencing data for individuals of African ancestry, biases in pathogenicity annotation may exist. Were any novel germline pathogenic variants identified in this cohort of African ancestry? This could represent a missed opportunity to uncover significant findings.
6. The bioinformatics pipeline for detecting somatic alterations between their cohort and the TCGA cohort is different, which requires additional quality control. For example, comparison of INDELs in Figure 5C could also be explained if their cohort used a more sensitive INDEL caller, which may lack biological significance.
7. The aging signature is particularly vulnerable to FFPE-induced artifacts. Analysis used aging signature should not be

over-interpreted.

8. While RNA-seq was performed, its application appears limited to fusion detection. Consider expanding its use could provide novel insights.

9. P-values are not consistently reported in the text. For example, the comparison of TP53 mutations between populations does not mention statistical results in the main text or Figure 2. Similarly, the survival analysis mentions hazard ratios (HR) but omits p-values.

Reviewer #1 (Remarks on figshare data availability):

Mutation calls and expression data are in figshare. Raw sequencing bam files, which are more important are being deposited to dbGap but can't be reviewed.

Reviewer #2 (Remarks to the Author):

A. Summary of the key results

Triple-negative breast cancer (TNBC) has a higher incidence in Black women and may reflect differences in cancer biology or socio-environmental exposure. Black women have been underrepresented in genomic studies and the authors have assembled a large collection (n= 513) of tumor and normal tissue from Black women with TNBC and performed DNA and RNA sequencing to determine if tumor genomic landscape of TNBC in Black women. They demonstrate that TP53 mutations are dominant (95% mutated), a low frequency of PIK3CA mutations and identify several potential driver genes unique to Black patients. The investigators show differing mutational signatures that associate with etiology and prognosis with one occurring in younger patients with deficient DNA repair and another in older patients and associated with aging and obesity. Overall, this is the largest TNBC genomic study in Black women and providing a better understanding of the genomics in an underrepresented population with disproportioned incidence. The authors demonstrate that the genomic landscape of somatic mutations is largely similar to white women with some minor differences.

B. Originality and significance:

The manuscript is original and addresses an important question as to whether genomics have a major role in the differences in incidence with Black women with TNBC.

C. Data & methodology:

The sequencing and mutation calling methods are excellent. Additional information is needed on how TNBC samples were identified from the varying cohorts.

D. Appropriate use of statistics and treatment of uncertainties:

Statistical methods are appropriate

E. Conclusions:

Additional mutational analysis from Non-hispanic white tumors would improve the robustness when making comparisons between the Black women cohort. Only 69 samples from the TCGA were used to evaluate differences in p53 mutation. There should be plenty of other cohorts(SCAN-B, metabric and other) to perform a more robust analysis.

F. Suggested improvements: experiments, data for possible revision

1. In Figure 2 the mutation frequencies for R175H, R273H and R248 appear quite low for TCGA. Also, the amino acid positions do not line up between TCGA and B-CAUSE (particularly R273H). Please double check the TCGA and make sure the same ensemble annotation was used to make the comparison. There are several isoforms of P53 and differing amino acid annotations at these positions. Often .vcf files give amino acid positions for multiple transcripts (ENST's). The differences in specific sites could also be a reflection of low sample size. The amount of TNBC patients in the TCGA analysis is quite low (69 non-Hispanic White TNBC). Perhaps the authors should make comparisons to SCAN-B and the Fudan TNBC cohorts as well.

2. Others have shown differencing enrichment of mutations among TNBC subtypes (PMID: 30853353, PMID: 38811720 and PMID: 25103565) such as an PIK3CA mutations in the LAR subtype. Was there any subtype-specific mutation enrichment in this cohort? Were the germline mutations evenly distributed among TNBC subtypes?

3. Were the mutational signatures differently enriched in the TNBCtype transcriptional subtypes?

4. Were there any mutations in known mismatch repair genes in the five hypermutated tumors?

5. The investigators defined SBS signature-based TNBC subtypes by combining SBS1&5 (aging) and SBS3 (HRD). Why was APOBEC subtype left out? What is the survival like in the APOBEC (SBS2&13) dominant samples? They then bin these into Subtype 1 (low aging and high HRD) and Subtype 3 (high aging and low HRD). Subtypes 2 and 4 were not adequately discussed. Are they (high aging and high HRD) and (low aging and low HRD)? Consistent subtype labeling in

Figure 4 would also help interpretation.

6. In Table S7 the authors perform a cox proportional hazard model adjusting for age and stage. Why is it that the hazard ratio increases with stage for SBS1&5 as expected and decreases with increasing stage for SBS3? Please include number of samples (n=) in each group for this analysis.

7. RNA fusions could be discussed in greater detail. The NTRK3-ETV6 fusion is a marker for secretory breast cancer. PTK2 has multiple fusions partners (there are six tumors with a fusion). Do they all preserve the kinase domain? There are several NOTCH fusions are they exclusive with notch pathway mutations. There are also three patients with PARG-BMS1 fusions.

8. The authors show mention tumors had copy number gain and high-level amplification of CD274 (PD-L1) were associated with higher mRNA expression. Please show mRNA levels in relation to CD274 copy number in supplemental data.

9. The investigators mention putative driver mutations in CACNA1H, BACH2, EPHB3, and ITGB4 TNBC in Black women, but do not provide many details. Including a lollipop plot showing the types of mutations and locations may be helpful and include a discussion about their potential impact (GOF, LOF).

10. Of note there were 12 patients with ERBB2 amplifications (Table S9) and caution should be considered when analyzing these, as they be false negatives from prior clinical assays used to determine HER2 amplifications. How were the patients from each cohort identified as TNBC? HER2 amplified could be identified as FISH positive or 3+ by IHC.

G. References: appropriate credit to previous work?

Lehmann et al (PMID: 34725325) is mentioned several times in the figure legends, but is absent from the references.

Clarity and context:

Clarity and context are appropriate.

Reviewer #2 (Remarks on figshare data availability):

The authors provide a zip file containing germline and somatic mutation calls, CNA call, mutational signatures and normalized RNA seq data. These data are sufficient for secondary analysis.

Version 1:

Decision Letter:

Our ref: NG-A67995R

22nd May 2025

Dear Dr Yao,

Thank you for submitting your revised manuscript "Mutational Landscape of Triple-Negative Breast Cancer in African American Women" (NG-A67995R). It has now been seen by the original referees and their comments are below. The reviewers find that the paper has improved in revision, and therefore we'll be happy in principle to publish it in Nature Genetics, pending minor revisions to satisfy the referees' final requests and to comply with our editorial and formatting guidelines.

Sincerely,

Safia Danovi, PhD
Senior Editor, Nature Genetics
ORCID: 0009-0007-7822-5479

Reviewer #1 (Remarks to the Author):

I appreciate the authors' thorough responses to my comments, including their efforts to correct potential false positives and incorporate additional data and analyses into the manuscript. While the removal of several findings (e.g. TERT mutations) may slightly reduce the novelty, the study remains a valuable resource for the field. One additional suggestion is to

emphasize the broader significance of studying breast cancer in African American women, particularly in light of current challenges to diversity in research. Highlighting this perspective can reinforce the importance of inclusive scientific efforts that benefit all populations.

Reviewer #2 (Remarks to the Author):

The authors have sufficiently addressed by concerns and improved the manuscript. However, the authors should provide a summary of the key findings in the first paragraph of the discussion(below) in the abstract.

“The findings reveal that the mutational landscape of TNBC in AA women is largely similar to that in Asian and NHW women and there was no evidence of associations of mutational features with African ancestry. Therefore, our results do not support major racial differences in TNBC biology at the level of somatic mutations.”

Reviewer #2 (Remarks on figshare data availability):

Secondary analysis provided in figshare. Could not verify that the raw data has been deposited in DBGaP (phs003962.v1.p1).

Response to Referees

We sincerely thank the editor and the two reviewers for their recognition of the quality and importance of our work. Moreover, we thank them for the insightful and constructive comments, which we took wholeheartedly and addressed to the fullest extent of our capability. Major changes in the revised manuscript include the following:

- 1) As advised by both reviewers, we expanded the comparison groups to 279 Asian TNBC patients and 626 Non-Hispanic White (NHW) patients by aggregating public available TNBC genomic data, for comparisons across the three populations.
- 2) We revised the significantly mutated gene (SMG) analysis by using MutSig2CV.
- 3) We performed in-depth pathogenicity analyses of germline variants by comparing our data to large reference population datasets, as well as cross-referencing recent functional characterization studies.
- 4) We conducted analyses of tumor mutations by TNBC transcriptional subtypes.
- 5) We expanded the analyses to better leverage the tumor transcriptomic data, including new gene set enrichment analysis and neoantigen prediction.
- 6) We clarified several technical and methodological issues, including the classification of TNBC, *ERBB2* amplification, the bioinformatic pipeline, FFPE artifacts, etc.

Overall, the main findings of our study remain the same and the many improvements made in response to reviewers' comments have meaningfully strengthened our work. In particular, the analysis of germline variants added new results on pathogenicity in populations of African ancestry. Below, we provide a point-to-point response to the reviewers' comments, with revisions to the manuscript marked in track changes.

Reviewer #1 (Remarks to the Author):

Yao et al conducted an important study of whole-exome sequencing on a large cohort of triple-negative breast cancer (TNBC) patients of African ancestry. This effort addresses the critical need to understand the genetic basis of a disease that disproportionately affects Black women. The paper is well-written, and the data analysis is comprehensive. The most interesting finding is that TP53 is mutated in 95% of TNBC with African ancestry. There are several issues should be addressed. My specific comments are below.

1. TERT coding mutations have not been demonstrated as cancer drivers in large-scale genomic studies. Do these mutations functionally impact TERT expression? Are there any hotspot mutations or are those mutations clustered in a functional domain, suggesting they may act as driver mutations? Additionally, why do these mutations appear significant only in MuSiC but not in MutSigCV? It may be worth running MutSig2CV, an advanced version of MutSigCV that accounts for additional confounding factors to resolve this discrepancy and rule out the possibility of passenger events.

Response: Following the reviewer’s suggestion, we re-ran the SMG analysis using MutSig2CV and compared the results with those from MutSigCV and MuSiC. The data of top genes are shown in the table below.

gene	MutSig2CV		MutSigCV		MuSiC	
	p	q	p	q	p	q
TP53	1.00E-16	1.89E-12	0	0	0	0
RB1	4.11E-15	3.87E-11	0	0	0	0
PTEN	2.25E-14	1.42E-10	0	0	ns	ns
BRCA1	3.20E-07	1.51E-03	1.53E-06	3.21E-03	5.43E-10	5.41E-08
AXIN1	1.18E-06	4.46E-03	1.93E-04	1.98E-01	ns	ns
PIK3CA	5.96E-06	1.87E-02	1.84E-09	4.97E-06	0	0
NF1	7.65E-06	2.06E-02	7.64E-08	1.80E-04	2.73E-08	9.79E-05
SS18L1	9.79E-06	2.31E-02	1.03E-02	1	ns	ns
ARID1A	2.26E-05	4.47E-02	9.28E-04	6.73E-01	ns	ns
MLL3	2.37E-05	4.47E-02	1	1	5.61E-06	1.01E-02
PIK3R1	3.19E-05	5.46E-02	1.34E-10	4.20E-07	1.93E-06	4.62E-03
MLL2	4.08E-05	6.42E-02	1	1	5.61E-06	1.01E-02
KDM6A	1.02E-03	7.36E-01	1.47E-05	2.78E-02	1.91E-04	1.24E-01
NOTCH2	1.40E-03	8.81E-01	5.08E-05	7.37E-02	2.02E-05	2.79E-02
CBFB	1.89E-03	1	1.65E-05	2.82E-02	ns	ns
BACH2	6.19E-03	1	3.03E-05	4.76E-02	ns	ns
NOTCH1	5.48E-02	1	7.69E-05	1.04E-01	0	0
TERT	6.08E-02	1	3.78E-02	1	5.79E-09	2.37E-05

Footnote: ns: non-significant

Although the results from MutSig2CV are largely consistent with those from MutSigCV, there are also some notable differences. For example, *MLL3* (*KMT2C*) and *MLL2* (*KMT2D*), two known TNBC genes, are missed by MutSigCV but identified by MutSig2CV and MuSiC. In the case of *KDM6A*, *NOTCH1*, and *NOTCH2*, another three known TNBC genes, both MuSiC and MutSigCV are more sensitive than MutSig2CV to identify them. To be conservative, we limited the final list of SMGs to those significant ($q \leq 0.2$ consistent with the cutoff used by MuSiC) by at least two of the three algorithms (marked with blue font in the table above), all 13 of which are all known TNBC genes as shown a new **Extended Data Table 2**. For future research reference, we provide the full results from MutSigCV, MutSig2CV, and MuSiC tests in **Supplementary Tables 2-4**.

For *TERT*, the SMG test is highly significant by MuSiC ($P = 5.8E-09$, $q = 2.4E-05$) but only nominally significant by MutSigCV ($P = 0.038$, $q = 1$) or MutSig2CV ($P = 0.061$, $q = 1$). We thus no longer classify it as an SMG based on the above criterion. Nevertheless, we confirmed the significantly higher frequency of coding mutations of *TERT* in our data (17 of 462 cases or 3.7% harboring aa-changing mutations, including two recurring mutations and one loss of function mutation), in comparison to Asian patients from FUSCC (3 of 279 cases harboring aa-changing mutations, or 1.1%) and NHW patients

from TCGA and SCAN-B (2 of 306 cases harboring aa-changing mutations, or 0.7%; *TERT* was not included in the METABRIC gene panel). We tested the impact of *TERT* aa-changing mutations with mRNA expression, which did not show significant difference ($P = 0.75$). Although we cannot rule out that some *TERT* coding mutations may affect the activity of telomerase reverse transcriptase as demonstrated in the setting of telomere biology diseases (PMID: 38641551), after synthesizing the above data and literature, we agree with the reviewer that these mutations could be passenger events. We have thus removed the mentioning of *TERT* and revised the Results section on Page 4 accordingly, which now reads:

“SMG analysis identified 13 genes with $q \leq 0.20$ by two or more programs used (MutSigCV²¹, MutSig2CV²², and MuSiC²³), which were all known cancer genes in TNBC (Extended Data Table 2; Supplementary Tables 2-4).”

2. The study cohort is substantially larger than the TCGA triple-negative cohort of non-Hispanic white samples, which may lead to confounding results due to the lack of appropriate controls. For instance, mutations in *GNAS* and *GATA3* are well-established breast cancer genes found across all populations. However, the statement in lines 324–326 could mislead readers into thinking these genes are exclusive to Black women. “Several genes were found to mutate at a frequency $\geq 2\%$ in Black women only (*GNAS*, *TERT*, *FGFR3*, *FGFR4*, *BACH2*, *FLT4*, *MN1*, *RECQL4*, and *GATA3*).”

Response: We agree with the reviewer that the small number of NHW TNBC patients in TCGA as a comparison group could have led to spurious results. Following both reviewers’ suggestion, we expanded the comparison groups to 279 Asian patients from FUSCC and 626 NHW patients pooled from TCGA ($n=69$), SCAN-B ($n=237$), and METABRIC ($n=320$). We re-analyzed mutation frequency across African American (AA), Asian, and NHW patients, with the new results of known breast cancer genes summarized in an updated **Figure 4** that is also shown below. The full comparison analysis results are provided in a new **Supplementary Table 6**.

Although the main findings remain the same that the mutational landscape of TNBC in AA women is similar to that in Asian and NHW women, and AA TNBC patients had higher frequencies of *TP53* and *NOTCH1* mutations but lower frequency of *PIK3CA* mutation than Asian and NHW TNBC patients, the new analyses provide some more nuanced data of the similarity and difference of somatic mutations across the three populations. First, the mutation frequencies of most breast cancer genes were more similar between Asian and NHW TNBC patients, both of which were notably different from AA patients. Second, most the previously reported genes exclusively mutated in AA cases at a frequency $\geq 2\%$ (*GNAS*, *FGFR3*, *FGFR4*, *BACH2*, *FLT4*, *MN1*, *RECQL4*, and *GATA3*) were due to small number of NHW patients as speculated by the reviewer. With a much larger NHW comparison group, although the frequencies for *FGFR3*, *FGFR4*, *BACH2*, *FLT4*, *RECQL4*, and *GATA3* in NHW patients were still very low ($< 1\%$), they (except for *RECQL4*) were not exclusive to AA. Third, several previously reported genes with much higher frequency in NHW than AA patients, including *HMCN1*, *ARID1B*, and *DMD*, now had a more similar frequency between the two

groups. Fourth, aside from *PIK3CA*, several other genes had a notably higher frequency in Asian or NHW patients than in AA patients, which included *ATR*, *RYR2*, *USH2A*, *MAP3K1*, *SETD2*, *AKT1*, *PREX2*, and *ATR* ($q < 0.1$).

a. African American (AA) vs. Non-Hispanic White (NHW)

b. AA vs. Asian

c. Asian vs. NHW

Figure legend: The frequency of non-silent mutations in known breast cancer genes in triple-negative breast cancer (TNBC) from a) African American (AA) women (y-axis) and non-Hispanic

White (NHW) women (x-axis); **b**) AA women (y-axis) and Asian women (x-axis); and **c**) Asian women (y-axis) vs. NHW women (x-axis). AA women were from B-CAUSE Study; Asian women were from FUSCC; and NHW women were pooled from TCGA, SCAN-B, and METABRIC. Each dot represents one gene with the dot size corresponding to the negated log₁₀-transformed p-value from comparison test. Genes that were significant at $q \leq 0.05$ after multi-testing correction by false discovery rate (FDR) are shown in red.

Based on these new data, we have revised the Results section on **Page 8**, which now reads:

“**Figure 4** shows the three-way comparisons of mutation frequency of known breast cancer genes across AA patients from B-CAUSE Study (n = 462), Asian patients from FUSCC (n = 279)¹⁰, and NHW patients (n = 626) pooled from TCGA³⁵, SCAN-B¹¹, and METABRIC³⁶, with the full results provided in **Supplementary Table 6**. The mutation frequencies were largely similar between Asian and NHW patients, yet several genes had notably different mutation frequency from these in AA patients, including higher frequencies of *TP53* (95%, 78%, and 75% in AA, Asian, and NHW patients, respectively; $P < 1E-09$) and *NOTCH1* (7%, 2%, and 4%, respectively; $P < 0.01$) mutations. On the contrary, a number of genes were found to mutate at a lower frequency in AA than Asian or NHW patients, most notably *PIK3CA* (5%, 19%, and 15%, respectively; $P < 2E-06$), *RYR2* (3%, 7%, and 8%, respectively; $P < 0.03$) and *USH2A* (3%, 1%, and 8%, respectively; $P < 9E-04$), while the mutation frequency of *AKT1*, *ATR*, *ATRX*, *MAP3K1*, *PREX2*, and *SETD2* were very low in AA patients (< 1%) but relatively common ($\geq 3\%$) in Asian and/or NHW patients.”

3. Similarly, TP53 mutation analysis is limited to comparing their cohort with TCGA, while including other existing cohorts (i.e. AACR GENIE) can add more controls to increase the power.

Response: Since AACR GENIE data (public release v17) does not provide annotations to properly classify TNBC, we instead used mutation data of 279 Asian patients from FUSCC and 626 NHW cases combined from TCGA, SCAN-B, and METABRIC. Based on the new comparison data, we re-analyzed the mutation spectrums of *TP53* and *PIK3CA* as the two genes with the largest mutation frequency difference but in opposite direction, across AA, Asian, and NHW TNBC patients, with the new results illustrated in an updated **Figures 5a** and **5b**, which are also shown below. The results demonstrate largely similar spectrums of *TP53* and *PIK3CA* mutations with some minor but notable differences across the three patient populations.

a. *TP53*

b. PIK3CA

Figure legend: Lollipop plots of a) TP53 and b) PIK3CA somatic mutations in triple-negative breast cancer (TNBC) from African American (AA) women in B-CAUSE, Asian women in

FUSCC, and Non-Hispanic White women pooled from TCGA, SCAN-B, and METABRIC. The numbers in the circles indicate the number of tumors harboring the mutation in the cohort.

Based on these new results, we have revised the Results section on **Page 8**, which reads:

“For *TP53* and *PIK3CA*, the two genes showing the largest mutation frequency discrepancy across the three patient populations, the gene mutation spectrums were, nevertheless, largely similar, with some minor yet notable differences. Most of the mutations in *TP53* were found in the DNA-binding domain (DBD), featured prominently with four hotspot mutations (R175, R213, R248, and R273), plus another hotspot mutation R342 in the tetramerization domain (**Figure 5a**). Tumors from AA women had two other hotspot mutations, H179 and E286 in the DBD, which were absent in Asian patients and at only low frequency in NHW patients. On the contrary, non-sense mutation R196* was rare in AA patients but more common in Asian and NHW patients. The spectrum of *PIK3CA* mutations was dominated by one hyperactivating mutation H1047R/L in all three populations; however, other three hyperactivating hotspot mutations, N345K, E542K, E545K, were found only in tumors from Asian and NWH but not AA women (**Figure 5b**).”

4. The survival analyses pool patients from multiple institutions and hospitals and analyze them together. Differences in neighborhood demographics and access to quality breast cancer care could confound the survival results. These disparities should be accounted for to ensure accurate conclusions. For example, survival analysis at different site should be analyzed separately, or meta-analyzed.

Response: This is an excellent point raised by the reviewer. We added study site as a confounder in the multivariable models, which did not substantially alter the results (**Extended Data Table 3**). Following the reviewer’s suggestion, we also performed meta-analyses across study sites as shown in the forest plot below. The associations are consistent between WCHS and VUMC, the two studies with larger sample size, but not so with BWHS, where the sample size is more limited. For the combined SBS subtype 4, the Cox model did not properly converge in the analysis within BWHS, resulting in unreliable estimates. Tests for heterogeneities across studies were all non-significant ($P > 0.05$). These results support that our findings are less likely confounded by inter-study heterogeneity. We have added the forest plots in a new **Extended Data Figure 4** and added one sentence in the Results section on **Page 8**, which reads:

“Meta-analyses across the three study sites show similar results (**Extended Data Figure 4**).”

Figure legend: The low, medium, and high levels of SBS1&5 (aging) and SBS3 (HRD) were determined by tertiles of the signatures. The low levels were used as the reference group (not shown in the figures). In the analysis of the combined SBS1&5 and SBS3 group, the Subtype 1 (low aging and high HRD) was used as the reference group (not shown in the figure). For Subtype 4, the model did not converge in BWHS and the results are not shown.

5. The criteria for defining pathogenic germline variants specifically for African ancestry patients are unclear. Given the lack of germline sequencing data for individuals of African ancestry, biases in pathogenicity annotation may exist. Were any novel germline pathogenic variants identified in this cohort of African ancestry? This could represent a missed opportunity to uncover significant findings.

Response: We fully agree with the reviewer that the classification of pathogenic variants in populations of African ancestry can be inaccurate due to the lack of proper reference data. Our original classification of pathogenic mutations includes frameshift, nonsense, and splice site variants, plus missense variants predicted to be “damaging or possibly damaging” on the protein structure and function by Polyphen.

To better leverage the germline sequencing data from a large population of AA TNBC patients in our study as advised by the reviewer, we have since re-analyzed the germline mutations in 9 known breast cancer genes, including *ATM*, *BRCA1*, *BRCA2*, *CDH1*, *CHEK2*, *NF1*, *PALB2*, *PTEN*, and *TP53*. A total of 124 unique mutations were found in 241 patients. The information of all these variants is now provided in a new **Supplementary Table 5**. We first queried these variants in gnomAD data of AFR ancestry, which has information on 105/124 of the variants. We also queried them in other sequencing datasets and found 5 additional variants in the Regeneron Genetics Center (RGC), 4 in the NIH All of US, and 1 in TOPMed. Notably, 22 of these 115 variants were found exclusively in populations of African ancestry.

We formally tested the frequency of these mutations in TNBC patients in B-CAUSE against the frequency in reference data from populations of African ancestry in the public datasets, with the results shown in a new **Extended Data Figure 3** that is also shown below. With one exception of a multi-nucleotide variant (MNV) in *PTEN*, all other 60 variants with $P < 0.05$ at a nominal significance level had a higher frequency in TNBC cases in B-CAUSE than in the AFR reference data.

Figure legend: The minor allele frequency in B-CAUSE data (x-axis) and AFR reference data (y-axis) are log-transformed and negated and germline variants are color coded according to ClinVar classification. P-values from Fisher's exact test are indicated by the dot diameter.

Of these 60 variants, 2 were originally classified as benign, 23 as pathogenic or likely pathogenic, 14 as conflicting classifications of pathogenicity, 15 as uncertain significance (VUS), and 6 variants not reported in ClinVar. Of the 22 variants found exclusively in populations of African ancestry, all but 3 variants had a significantly higher frequency in B-CAUSE TNBC patients than in the reference AFR datasets ($P < 0.05$).

We then looked up the germline variants identified in our dataset in recent functional characterization studies of *BRCA1* (Findlay 2018 Nature PMID: 30209399), *BRCA2* (Huang 2025 Nature PMID: 39779857), *CHEK2* (McCarthy-Leo 2024 PLOS Genet PMID: 39146382), and *TP53* (Funk 2025 Nat Genet PMID: 39774325). We were able to identify 18 variants with functional data in these experimental studies, 8 of which were found to be damaging. The details are shown in **Supplementary Table 5**.

When synthesizing the above results, we were able to validate the accuracy of ClinVar's pathogenicity classification for most germline variants in our study. Specifically, our data

supported the benignity classification (benign/likely benign) for most variants: only 2 out of 30 variants showing a significantly higher frequency ($P < 0.05$) in B-CAUSE TNBC cases than in the reference AFR datasets, and both were found non-damaging in functional studies. Similarly, our data confirmed the pathogenicity classification (pathogenic/likely pathogenic) for most variants: 23 out of 25 variants showed a significantly higher frequency ($P < 0.05$) in B-CAUSE TNBC cohort than in the reference AFR datasets, and all 4 variants evaluated in functional studies were found to be damaging.

For variants with conflicting or ambiguous classification in ClinVar, our data provide new evidence to refine the pathogenicity classification. Of the 35 variants with “conflicting classification of pathogenicity” and 18 VUS in ClinVar, 14 and 15 had a higher frequency in B-CAUSE TNBC patients than in the reference AFR datasets, respectively ($P < 0.05$). Further, three of the variants were assessed in functional studies and two were found to be damaging.

We identified six variants, including two in *PTEN* and four in *BRCA1*, that receive no pathogenicity classification in ClinVar. All six variants have a higher frequency in B-CAUSE TNBC patients than in the reference AFR datasets ($P < 0.05$), including R119C in *PTEN* found in three TNBC patients ($P = 9E-7$). Lastly, we discovered nine variants in our study, including three in *BRCA1*, three in *BRCA2*, two in *PALB2*, and one in *NF1*, which are not reported in any population databases. Two of the *BRCA1* variants were determined to be damaging in saturation genome editing (PMID: 30209399).

Detailed information of all the germline variants is provided in **Supplementary Table 5**. The raw germline sequencing data have been deposited to dbGaP (pfs003962.v1.p1). Based on these results, we have revised the Results section on **Page 5**, which reads:

“Using sequencing data from matched normal samples, we identified 124 germline mutations in nine known TNBC predisposition genes from 241 patients (**Figure 1a**). Of these variants, 115 were found in gnomAD and other reference datasets²⁴⁻²⁷, 22 being exclusive to populations of African ancestry (**Supplementary Table 5**). When minor allele frequency was compared to reference populations, 60 variants had a higher frequency in TNBC patients ($P < 0.05$) (**Extended Data Figure 3**). These results confirmed benignity for 28 of 30 variants classified as “benign/likely benign” and pathogenicity for 23 of 25 variants classified as “pathogenic/likely pathogenic” by ClinVar²⁸, while yielding new evidence of pathogenicity for 14 of 35 variants annotated as “conflicting classification of pathogenicity” and 15 of 18 variants annotated as “uncertain significance”. Moreover, we identified six variants, including two in *PTEN* and four in *BRCA1* with no pathogenicity annotation in ClinVar²⁸, all of which had a higher frequency in TNBC cases than in the reference datasets, including R119C mutation in *PTEN* ($P = 9E-07$). Lastly, we discovered nine novel germline mutations not previously reported in any reference databases, including three in *BRCA1*, three in *BRCA2*, two in *PALB2*, and one in *NF1*. Two of the *BRCA1* variants were deemed damaging in saturation genome editing²⁹.”

6. The bioinformatics pipeline for detecting somatic alterations between their cohort and the TCGA cohort is different, which requires additional quality control. For example, comparison of INDELS in Figure 5C could also be explained if their cohort used a more sensitive INDEL caller, which may lack biological significance.

Response: The downstream of our bioinformatics pipeline includes an additional QC step by meticulous manual review. Further, to tease out whether the high *TP53* mutation frequency in our data was real or due to technical artifacts, we carried out two orthogonal replication analyses using RNAseq data and targeted amplicon resequencing. Both analyses confirmed essentially all *TP53* mutations initially identified in whole-exome sequencing data (**Extended Data Figure 2**). Thus, the high frequency of *TP53* mutations in our TNBC data is unlikely a technical artifact. Lastly, we concur with the reviewer and explicitly acknowledge possible technical differences between our data and other data in the Discussion section on Page 10, which reads:

“The difference could be due to TNBC risk factors more common in AA women, or alternatively, to technical differences in sequencing or variant identification. Compared to TCGA and other WES-based studies, we used a larger exome library design with sizable custom contents, longer 150-bp reads, and deeper sequencing depth. Variant calling algorithms could have also contributed to the difference in mutation frequency, where we integrated four callers followed by a manual review step for quality control purpose. The fact that we validated essentially all *TP53* mutations in two orthogonal replication analyses supported the internal validity of our variant calling. However, caveats should be taken when comparing our data with those from external studies for these technical differences.”

7. The aging signature is particularly vulnerable to FFPE-induced artifacts. Analysis used aging signature should not be over-interpreted.

Response: This is an excellent point that warrants careful consideration.

First, we assessed the correlation between the two aging signatures SBS1 and SBS5 and chronological age in our data derived from FFPE tissues. The correlation (Spearman $r = 0.21$) is on par with these from fresh frozen samples in TCGA or SCAN-B data (Spearman r 's = 0.13-0.34), suggesting that the aging signal in our data is not unusually inflated.

Second, we considered the biochemical origin of FFPE artifacts in relation to sequencing library preparation protocols. To remedy the deamination artifacts in FFPE samples, some sequencing library prep protocols incorporate a repair treatment step using uracil DNA glycosylase (UDG) to remove uracil bases prior to amplification. During this process, formalin-induced deamination of 5-methylcytosine in CG dinucleotides is converted to thymine instead of uracil, which cannot be corrected by the UDG treatment. This artifactual FFPE-signature after UDG treatment could mimic the aging signature SBS1. In contrast, in FFPE samples not treated by UDG prior to sequencing, the artifact is similar to SBS30 that is attributed to defects in base excision

repair (PMID: 36068219). Because we did not perform UDG treatment in our library prep workflow, the FFPE-induced artifacts in our data, when present, would presumably result in a signature more similar to SBS30, which we did not detect in our data. This suggests to us that contamination of the SBS1 aging signature by FFPE artifacts might not be pervasive in our data.

To address this concern explicitly, we have added a caveat about the overlap between FFPE artifacts and aging signature in the Discussion section on Page 10, which reads:

“It is known that SBS1 is similar to FFPE artifacts after chemical repairment in DNA library preparation⁵¹. Although we did not perform the repairment step, cautions are advised when interpreting the results related to SBS1 signature.”

8. While RNA-seq was performed, its application appears limited to fusion detection. Consider expanding its use could provide novel insights.

Response: Thanks for this constructive suggestion! In addition to fusion detection, RNA-seq data were also used for TNBC transcriptional subtyping (**Figure 1 and Supplementary Figure 3**), splicing analysis of noncoding mutations of *TP53* (**Extended Data Figure 1**), validation of *TP53* mutations (**Extended Data Figure 2**), and immune signature analysis (**Figure 4c**).

Following the reviewer’s suggestion, we expanded the analyses of RNA-seq data to differential expression analysis by SBS signature-based subtype followed by gene set enrichment analysis (GSEA) (**Supplementary Figure 7**) and neoantigen analysis (**Supplementary Figure 10**). Interestingly, GSEA revealed significant enrichment of many immune response pathways in Subtype 1 (low aging and high HRD) relative to Subtype 3 (high aging and low HRD), consistent with those shown in **Figure 4c**. We added one sentence on this in the Results section on Page 7, which reads:

“Consistently, GSEA showed significant enrichment of many immune response gene sets in Subtype 1 relative to Subtype 3 tumors (Supplementary Figure 7).”

9. P-values are not consistently reported in the text. For example, the comparison of TP53 mutations between populations does not mention statistical results in the main text or Figure 2. Similarly, the survival analysis mentions hazard ratios (HR) but omits p-values.

Response: Thanks for pointing this out. We have made revisions to report p-values consistently throughout the manuscript.

Reviewer #1 (Remarks on figshare data availability):

Mutation calls and expression data are in figshare. Raw sequencing bam files, which are more important are being deposited to dbGap but can't be reviewed.

Response: dbGaP submission of the bam files from 462 tumor-normal pairs under accession number phs003962.v1.p1 is currently pending final release by the NCBI team.

Reviewer #2 (Remarks to the Author):

A. Summary of the key results

Triple-negative breast cancer (TNBC) has a higher incidence in Black women and may reflect differences in cancer biology or socio-environmental exposure. Black women have been underrepresented in genomic studies and the authors have assembled a large collection (n= 513) of tumor and normal tissue from Black women with TNBC and performed DNA and RNA sequencing to determine if tumor genomic landscape of TNBC in Black women. They demonstrate that TP53 mutations are dominant (95% mutated), a low frequency of PIK3CA mutations and identify several potential driver genes unique to Black patients. The investigators show differing mutational signatures that associate with etiology and prognosis with one occurring in younger patients with deficient DNA repair and another in older patients and associated with aging and obesity. Overall, this is the largest TNBC genomic study in Black women and providing a better understanding of the genomics in an underrepresented population with disproportioned incidence. The authors demonstrate that the genomic landscape of somatic mutations is largely similar to white women with some minor differences.

B. Originality and significance:

The manuscript is original and addresses an important question as to whether genomics have a major role in the differences in incidence with Black women with TNBC.

C. Data & methodology:

The sequencing and mutation calling methods are excellent. Additional information is needed on how TNBC samples were identified from the varying cohorts.

Response: We are grateful for the positive feedback and thoughtful summary of our study. TNBC patients were identified as patients with invasive primary breast cancer with ER, PR, ERBB2 (HER2) all being negative based on pathology reports at diagnosis that were collected along with tumor tissues. Tumor ER and PR status was determined by immunohistochemical (IHC) staining. Tumor HER2 status was determined by IHC staining, complemented with fluorescence in-situ hybridization (FISH) for those with equivocal IHC results (2+). This classification is consistent with the approach used in the parent studies.

D. Appropriate use of statistics and treatment of uncertainties:

Statistical methods are appropriate

E. Conclusions:

Additional mutational analysis from Non-hispanic white tumors would improve the robustness when making comparisons between the Black women cohort. Only 69

samples from the TCGA were used to evaluate differences in p53 mutation. There should be plenty of other cohorts (SCAN-B, metabric and other) to perform a more robust analysis.

Response: Following both reviewers' suggestion, we have expanded the number of TNBC patients from NHW women by aggregating data from TCGA, METABRIC and SCAN-B and added TNBC cases from Asian women in FUSCC. Please refer to the response to Reviewer 1 comment #2 above for details.

F. Suggested improvements: experiments, data for possible revision

1. In Figure 2 the mutation frequencies for R175H, R273H and R248 appear quite low for TCGA. Also, the amino acid positions do not line up between TCGA and B-CAUSE (particularly R273H). Please double check the TCGA and make sure the same ensemble annotation was used to make the comparison. There are several isoforms of P53 and differing amino acid annotations at these positions. Often .vcf files give amino acid positions for multiple transcripts (ENST's). The differences in specific sites could also be a reflection of low sample size. The amount of TNBC patients in the TCGA analysis is quite low (69 non-Hispanic White TNBC). Perhaps the authors should make comparisons to SCAN-B and the Fudan TNBC cohorts as well.

Response: Following both reviewers' suggestion, we have expanded the comparison groups to 279 Asian TNBC patients from FUSCC and 626 NHW TNBC patients pooled from TCGA, SCAN-B, and METABRIC. We re-reviewed the ensemble annotation of *TP53* mutations and the alignment on the lollipop plots to ensure consistency. All mutations were mapped using the same Ensembl transcript reference (ENST00000269305, *TP53* canonical transcript) to ensure proper alignment of amino acid positions across datasets. Upon re-inspection, we confirmed that the alignment of R273H and other hotspot mutations was correct in the original plot. With the expanded analyses across three populations, we now show the mutational spectrum in a separate panel for each population. Similarly to *TP53*, we also revised the mutational spectrum lollipop plots for *PIK3CA*. Please refer to our response and the figures to Reviewer 1's comment #3 above. Note we have renumbered this figure to **Figure 5**.

2. Others have shown differencing enrichment of mutations among TNBC subtypes (PMID: 30853353, PMID: 38811720 and PMID: 25103565) such as an *PIK3CA* mutations in the LAR subtype. Was there any subtype-specific mutation enrichment in this cohort? Were the germline mutations evenly distributed among TNBC subtypes?

Response: We compared the frequency of germline and somatic mutations across TNBC transcriptional subtypes, which showed an enrichment of somatic mutations in *PTEN* ($P = 0.003$) and *PIK3R1* ($P = 0.04$) and depletion of *TP53* ($P = 0.009$) mutation in the LAR subtype, consistent with these shown in previous studies. No significant difference was found for germline mutations. These results are now shown in **Supplementary Figure 3** and a brief description is added to the Results section on **Page 5**, which reads:

“In comparisons across TNBC transcriptional subtypes^{17,18}, the LAR subtype had enrichment of somatic mutations in *PTEN* ($P = 0.003$) and *PIK3R1* ($P = 0.04$) and slight depletion of *TP53* ($P = 0.009$) mutation, consistent with previous studies in Asian and NHW patients (**Supplementary Figure 3**)^{10,19,20}”

Figure legend: Comut plot of somatic and germline mutations in triple-negative breast cancer (TNBC) from AA women. Mutation rate (first panel) is presented as number of SNVs per mb. Proportion African ancestry (second panel) was estimated based on germline variant data from matched normal DNA samples and presented as a numeric value between 0 and 1. Homologous recombination deficiency (HRD) (third panel) was estimated based on whole exome sequencing data using scarHRD R package (PMID: 29978035). TNBC subtype (fourth panel) was classified based on tumor transcriptomic data available from 260 of the cases using the method by Lehmann et al (PMID: 34725325). BL1: basal-like 1; BL2: basal-like 2; LAR: luminal androgen receptor; M: mesenchymal; UNS: unassigned. Somatic mutations of selected genes (fifth panel) are sorted by mutation frequency. Germline variants of *BRCA1* and *BRCA2* are shown in the sixth panel. Insertion/deletion (ID) mutational signatures and single base substitution (SBS) signatures are shown in the seventh and eighth panel, respectively.

3. Were the mutational signatures differently enriched in the TNBCtype transcriptional subtypes?

Response: Following this Reviewer’s suggestion, we compared SBS and ID signatures across TNBC transcriptional subtypes. The results are combined with these from comparisons of somatic and germline mutations as describe above and shown in **Supplementary Figure 3**. We noted relatively lower APOBEC-related SBS2 ($P = 0.005$) and SBS13 ($P = 0.09$) in the M subtype, and relatively lower HRD-related SBS3 ($P = 0.006$) and higher ID4 signature ($P = 0.006$) without known etiology in the LAR subtype. A brief description of these results is added to the Results section on **Page 7**, which reads:

“When examined across TNBC transcriptional subtypes^{17,18}, the M subtype had relatively lower APOBEC-related SBS2 ($P = 0.005$) and SBS13 ($P = 0.09$) signatures, and the LAR subtype had lower HRD-related SBS3 ($P = 0.006$) but higher ID4 signature ($P = 0.006$) (**Supplementary Figure 4**).”

4. Were there any mutations in known mismatch repair genes in the five hypermutated tumors?

Response: We queried a panel of 22 mismatch repair (MMR) genes in the five hypermutated TNBC tumors, three of which harbored somatic mutations in MMR genes, one in each of *MLH1* [E319K], *MSH3* [E1018fs], and *LIG1* [I854M]. No germline mutations in any known breast cancer genes were found in these patients. We added this to the Results section on Page 3, which reads:

“with 5 tumors (1%) considered hypermutated (> 10 SNVs per Mb)¹³ (Supplementary Figure 2), including three carrying a mutation in mismatch repair genes (*MLH1*, *MSH3*, and *LIG1*).”

5. The investigators defined SBS signature-based TNBC subtypes by combining SBS1&5 (aging) and SBS3 (HRD). Why was APOBEC subtype left out? What is the survival like in the APOBEC (SBS2&13) dominant samples? They then bin these into Subtype 1 (low aging and high HRD) and Subtype 3 (high aging and low HRD). Subtypes 2 and 4 were not adequately discussed. Are they (high aging and high HRD) and (low aging and low HRD)? Consistent subtype labeling in Figure 4 would also help interpretation.

Response: Thanks for pointing this out. The two APOBEC-related SBS signatures together presented in 35% of the samples at a minor proportion relative to the other three SBS signatures, which could be considered dominant (accounting for ≥ 50% of the SNV mutations) in only 17 (5.3%) samples. Test of the combined SBS2 and SBS13 with survival outcomes did not yield significant results (SBS3&13 yes vs. no: overall survival HR = 0.72 [0.45-1.14] after adjustment for age, study, and stage). These were the reasons we did not focus on the APOBEC-related signatures.

For the SBS subtypes based on a combination of SBS1&5 (aging) and SBS3 (HRD), as the reviewer correctly speculated, Subtype 2 is low aging and low HRD, and Subtype 4 is high aging and high HRD. Neither was in significant association with patient survival in comparison to Subtype 1 (low aging and high HRD) as shown in **Extended Data Table 3**. We have since clarified this in the Results section on Page 8 and relabeled the plots in **Figure 3** to make it easier to distinguish the SBS-based subtypes. The three sentences added to the Results section read:

“The differences between Subtype 1 (low aging and high HRD) and Subtype 3 (high aging and low HRD) were the most apparent, whereas Subtypes 2 (low aging and low HRD) and Subtype 4 (high aging and high HRD) were somewhere in between.”

“No significant association of patient survival was found with APOBEC-related signatures SBS2 or SBS13.”

“No significant association was observed with Subtype 2 or Subtype 4.”

6. In Table S7 the authors perform a cox proportional hazard model adjusting for age and stage. Why is it that the hazard ratio increases with stage for SBS1&5 as expected and decreases with increasing stage for SBS3? Please include number of samples (n=) in each group for this analysis.

Response: We apologize for the confusion over this table. T1-T3 for SBS1&5 and SBS3 are not for tumor T stage, but the low, medium and high levels of the two mutational signatures based on the tertiles (T) of the data distribution. Tumor stage was adjusted for in the multivariable models, but we did not show the HRs associated with tumor stage. We double checked the univariate associations of cancer stage with patient survival, where higher stage was associated with poorer survival as expected. To avoid confusion, we have revised the labeling of the levels, added a footnote to clarify this, and added the number of patients in each subgroup as advised by the reviewer. Please also note this table is now referred to as **Extended Data Table 3**.

7. RNA fusions could be discussed in greater detail. The NTRK3-ETV6 fusion is a marker for secretory breast cancer. PTK2 has multiple fusion partners (there are six tumors with a fusion). Do they all preserve the kinase domain? There are several NOTCH fusions. Are they exclusive with notch pathway mutations? There are also three patients with PARG-BMS1 fusions.

Response: We thank the reviewer for these constructive suggestions. We have added discussion of these fusion events in the Results section on Page 9 and generated fusion plots as shown in **Supplementary Figure 11**, which can be seen in the next page.

Figure legend: Plots of fusion mutations involving NTRK3 and PTK2 detected in triple-negative breast cancer from AA women in B-CAUSE using transcriptomic data. The schemes on the right show the retained functional domains of the fusion proteins. The newly added result of fusion mutations reads:

“We characterized fusion events in 260 TNBC patients with transcriptomic data. Using stringent filtering criteria, we identified 471 fusion mutations in 148 (56%) of the tumors, including seven recurrent fusions and 96 fusions involving a known cancer gene (**Supplementary Table 7**). The most common recurrent fusions were characterized by adjacent rearrangements involving *PTK2* or *ETV6*, the latter of which is a tumor suppressor that turns to an oncogene in its fusion forms^{38,39}. We identified one tumor with *BCL2L14-ETV6* associated with mesenchymal TNBC⁴⁰ and another with *ETV6-NTRK3* that was a marker of secretory breast carcinoma, a rare basal-like breast cancer^{41,42}. Six tumors had fusions involving *PTK2* with multiple partners, and none retained the kinase domain (**Supplementary Figure 11**). Moreover, three tumors contained *PARG-BMS1* fusion associated with metaplastic TNBC⁴³. In addition, four tumors had fusion mutations involving *NOTCH2* or *NOTCH2NL*⁴⁴.”

8. The authors show mention tumors had copy number gain and high-level amplification of CD274 (PD-L1) were associated with higher mRNA expression. Please show mRNA levels in relation to CD274 copy number in supplemental data.

Response: To address this comment, we added a boxplot of mRNA expression with copy number call of CD274 in **Supplementary Figure 12**, which is also shown below. It confirms significantly higher expression in those with high-level amplification. We added one sentence to the Results section on Page 9, which reads:

“Lastly, 163 (35%) and 62 (13%) of the tumors had copy number gain and high-level amplification of *CD274* (*PD-L1*), respectively, associated with higher mRNA expression (**Supplementary Figure 12**)”

Figure legend: Boxplot of mRNA expression of CD274 (PD-L1) by CD274 copy number aberrations in 260 patients with tumor transcriptomic data in triple-negative breast cancer in AA

women from B-CAUSE. For copy number calls, 0 is neutral; -2 and -1 indicate homologous deletion and single-copy deletion, respectively; 1 and 2 indicate low-level and high-level amplification, respectively. The horizontal bar indicates the subgroup median, and the top and bottom of the box indicate the 75% and 25% percentile. P-value was derived from ANOVA.

9. The investigators mention putative driver mutations in CACNA1H, BACH2, EPHB3, and ITGB4 TNBC in Black women, but do not provide many details. Including a lollipop plot showing the types of mutations and locations may be helpful and include a discussion about their potential impact (GOF, LOF).

Response: In response to Reviewer 1's comment above, we re-ran the SMG analysis using MutSig2CV, in addition to MutSigCV and MuSiC as we previously did. Given the differences in the results from the three algorithms, we applied a conservative criterion to call a gene significantly mutated only if it is significant by at least two of the three algorithms ($q \leq 0.20$). Based on this criterion, none of the four genes is classified as an SMG. Please refer to the response to Reviewer 1's comment #1 for details. We have thus removed the mentioning of these four genes from the Results section. For future reference, we provide the full results of the SMG test using MutSig2CV, MutSigCV, and MuSiC in **Supplementary Tables 2-4**. For review purpose, the lollipop plots of these four genes (*CACNA1H*, *BACH2*, *EPHB3*, and *ITGB4*) are shown in the figure below.

Figure legend: Lollipop plots of *CACNA1H*, *BACH2*, *EPHB3*, and *ITGB4* somatic mutations in triple-negative breast cancer (TNBC) from AA women in B-CAUSE. Hotspot mutations are labeled with corresponding amino acid changes.

10. Of note there were 12 patients with ERBB2 amplifications (Table S9) and caution should be considered when analyzing these, as they be false negatives from prior clinical assays used to determine HER2 amplifications. How were the patients from each cohort identified as TNBC? HER2 amplified could be identified as FISH positive or 3+ by IHC.

Response: Thanks for pointing this out. Please refer to our response to comment C above for identification of TNBC cases. We reviewed published studies on *ERBB2* amplification in TNBC or basal-like subtype. As shown in Figure 1 of the 2012 landmark Nature paper on breast cancer genomic portrait from TCGA (PMID: 23000897), some basal-like patients did have *ERBB2* amplification. In the FUSCC-TNBC dataset, 10 out of 277 patients had *ERBB2* amplification based on CNV results (PMID: 30853353). Similarly, in METABRIC dataset 11 out 320 TNBC patients had *ERBB2* amplification based on CNV results (PMID: 27161491). This discordance between clinical HER2 assay and CNV analysis could be due to false negative test from clinical assays as the reviewer suggested, or it could be due to inaccurate CNV analysis, or tumor heterogeneity between samples used for clinical assays and samples used later for research analysis. It has also been shown that some TNBC tumors have activated ERBB2 signaling (PMID: 30853353). We added a blurb about this in the Discussion section on **Page 11**, which reads:

“Notably, 12 TNBC patients had *ERBB2* amplification despite being HER2 negative in clinical assays, which was also found in other TNBC studies^{10,35,36}. This discordance could be due to false negative clinical test, false positive CNV results, or tumor heterogeneity between samples used for clinical assays and these used later for research analysis. It might also reflect biological heterogeneity of TNBC as it has been shown that some TNBC tumors had activating ERBB2 signaling¹⁰.”

G. References: appropriate credit to previous work?

Lehmann et al (PMID: 34725325) is mentioned several times in the figure legends, but is absent from the references.

Response: Thanks for pointing out this inadvertent omission! We have added two references #17 and #18.

Clarity and context:

Clarity and context are appropriate.

Reviewer #2 (Remarks on figshare data availability):

The authors provide a zip file containing germline and somatic mutation calls, CNA call, mutational signatures and normalized RNA seq data. These data are sufficient for secondary analysis.

Response: dbGaP submission of the bam files from 462 tumor-normal pairs under accession number phs003962.v1.p1 is currently pending final release by the NCBI team.

Response to Referees

We thank the Editor and the two Reviewers for carefully reviewing our revised work and are encouraged by their positive feedback. Below is a point-to-point response to their remaining comments.

REVIEWER #1 COMMENTS:

I appreciate the authors' thorough responses to my comments, including their efforts to correct potential false positives and incorporate additional data and analyses into the manuscript. While the removal of several findings (e.g. TERT mutations) may slightly reduce the novelty, the study remains a valuable resource for the field. One additional suggestion is to emphasize the broader significance of studying breast cancer in African American women, particularly in light of current challenges to diversity in research. Highlighting this perspective can reinforce the importance of inclusive scientific efforts that benefit all populations.

Response: We added the following sentence to the conclusive paragraph in the Discussion section: *"The findings provide new insights into the disease epidemiology, etiology, and therapeutic vulnerability of TNBC from African American women that also deepens our understanding of this aggressive disease in other populations. Our research highlights the importance of continuous inclusive scientific efforts to ensure our rapidly growing cancer knowledge benefits all human populations."*

REVIEWER #2 COMMENTS:

The authors have sufficiently addressed by concerns and improved the manuscript. However, the authors should provide a summary of the key findings in the first Paragraph of the discussion(below) in the abstract. "The findings reveal that the mutational landscape of TNBC in AA women is largely similar to that in Asian and NHW women and there was no evidence of associations of mutational features with African ancestry. Therefore, our results do not support major racial differences in TNBC biology at the level of somatic mutations."

Response: Following this suggestion, we revised the abstract, which now reads: *"We unveiled a high-resolution mutational portrait of TNBC in African American women reminiscent of that in Asian and Non-Hispanic White women, with no evidence of associations of mutational features with African ancestry. We also made some distinctive discoveries, including an almost complete dominance of TP53 mutations, low frequency of PIK3CA mutations, and mutational signature-based subtypes with etiologic and prognostic significance. These findings do not support major racial differences in TNBC biology at the level of somatic mutations and contribute considerably to the diversity of the knowledgebase of breast cancer genomics and provide novel insights into the disease etiology, disparities, and therapeutic vulnerability of TNBC in African American women."*